# Understanding and Mitigating Robust Overfitting through the Lens of Feature Dynamics

## Abstract

Adversarial Training (AT) has become arguably the state-of-the-art algorithm for extracting robust features. However, researchers recently notice that AT suffers from severe robust overfitting problems, particularly after the learning rate (LR) decay, while the existing static view of feature robustness fails to explain this phenomenon. In this paper, we propose a new *dynamic* feature robustness framework which takes the dynamic interplay between the model trainer and the attacker into consideration. By tracing temporal and dataset-specific feature robustness, we develop a new understanding of robust overfitting from the dynamics of non-robust features, and empirically verify it on real-world datasets. Built upon this understanding, we explore three techniques to restore the balance between the model trainer and the attacker, and show that they could effectively alleviate robust overfitting and attain state-of-the-art robustness on benchmark datasets. Notably, different from previous studies, our interpretation highlights the necessity of considering the min-max nature of AT for robust overfitting.

## 1 Introduction

Overfitting seems to have become a history in the deep learning era. Contrary to the traditional belief in statistical learning theory that large hypothesis class will lead to overfitting, Zhang et al. (2019) note that DNNs have good generalization ability on test data even if they are capable of memorizing random training labels. Nowadays, large-scale training often does not require early stopping, and longer training simply brings better generalization (Hoffer et al., 2017).

However, researchers recently notice that in Adversarial Training (AT), overfitting is still a severe issue on both small and large scale data and models (Rice et al., 2020). AT is arguably the most effective defense method (Athalye et al., 2018) against adversarially crafted perturbations to images (Szegedy et al., 2014). Specifically, given training data $\mathcal{D}_{\text{train}}$ and model $f_\theta$, AT can be formulated as a min-max optimization problem (Madry et al., 2018; Goodfellow et al., 2015):

$$\min_\theta \mathbb{E}_{\bar{x},y \sim \mathcal{D}_{\text{train}}} \max_{x \in \mathcal{E}_p(\bar{x})} \ell_{\text{CE}}(f_\theta(x), y), \tag{1}$$

where $\ell_{\text{CE}}$ denotes the cross entropy (CE) loss, $\mathcal{E}_p(\bar{x}) = \{x \mid \|x - \bar{x}\|_p \le \varepsilon\}$ denotes the $\ell_p$-norm ball with radius $\varepsilon$. However, in practice, researchers notice this min-max training scheme suffers severely from the robust overfitting (RO) problem: after a particular point (*e.g.,* learning rate decay), its training robustness will keep increasing (Figure 1a, red line) while its test robustness will begin to dramatically decrease (Figure 1b, red line). This abnormal behavior of AT has attracted many interests recently. Previous work correctly points out that during AT, the robust loss landscape becomes much sharper, and RO can be largely surpassed by enforcing a smoother landscape with additional regularizations (Stutz et al., 2021; Chen et al., 2021; Wu et al., 2020). However, this phenomenological perspective does not explain the rise of either the sharp landscape or RO. Some researchers try to explain RO though the phenomenons of standard training (ST), such as random memorization (Dong et al., 2022) and double descent Dong et al. (2021), but they fail to characterize the uniqueness of AT from ST (why AT has the overfitting issue while ST does not).

In this paper, we seek to establish a generic explanation for robust overfitting during AT. Different from previous attempts to relate the behaviors of AT to existing theories of ST, we believe that the reasons of RO should lie exactly in the differences between AT and ST. In particular, we emphasize a critical difference: RO usually happens after learning rate (LR) decay in AT, while in ST, LR decay does not lead to (severe) overfitting, so our paper focus on the change of learning behaviors before and after the LR decay. We notice that feature robustness actually changes during this process: a non-robust feature could become robust after LR decays, which is contrary to the static feature robustness framework developed by Ilyas et al. (2019). This motivates us to design a *dynamic* feature

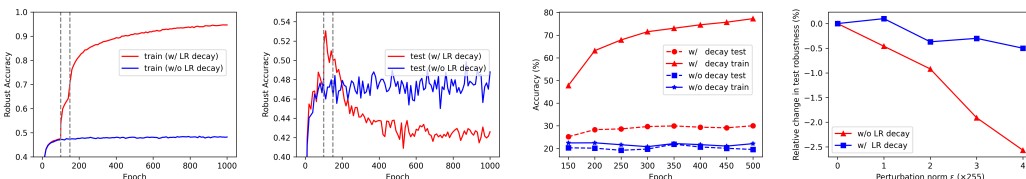

(a) training robustness during adversarial training (b) test robustness during adversarial training (c) robustness change of non-robust features (d) relative change in test robustness with stronger non-robust features ($\varepsilon$).

Figure 1: (a, b) Adversarial training results on CIFAR-10 using vanilla PGD-AT (Madry et al., 2018). (c) After decay, non-robust features become significantly more robust on training set, but only increase slightly on test set. (d) With LR decay, injecting stronger non-robust features (larger $\varepsilon$) induces severer degradation in test robustness, while the degradation is less obvious without decay.

robustness framework that takes the learning process into consideration, and discussion the dynamics of feature robustness. Moreover, this dynamic perspective also suggests a new understanding of robust overfitting. Specifically, we hypothesize that due to the strong local fitting ability endowed by smaller LR, the model learns false mapping of non-robust features contained in adversarial examples, which opens shortcuts for the test-time attacker and induces large test robust error. Accordingly, we design a series of experiments that empirically verify this understanding.

Based on this dynamic view of feature robustness, we investigate three new strategies to avoid fitting non-robust features: stronger model regularization, smaller LR decay, and stronger attacker. We show that all three strategies help mitigate robust overfitting to a large extent. Based on these insights, we propose Bootstrapped Adversarial Training (BoAT) that has neglectable (if any) degree of robust overfitting without using additional data augmentations, even if we train for 500 epochs. Meanwhile, BoAT attains state-of-the-art robustness (both best-epoch and final-epoch) on benchmark datasets including CIFAR-10, CIFAR-100, and Tiny-ImageNet. This suggests that with appropriate design of training strategies, AT could also enjoy similar "train longer, generalize better" property like ST. To summarize, our main contributions are:

- We point out that the existing static robust feature framework fails to explain robust overfitting, particularly after the LR decay. Therefore, we propose a new *dynamic* robust feature framework to account for this change by taking the interplay among the model trainer and the attacker into consideration.
- Based on our dynamic framework, we analyze the change of feature robustness during LR decay. We further propose a new understanding of robust overfitting through the change of non-robust features, and empirically verify it on three nontrivial implications.
- From our dynamic perspective, we propose three effective approaches that mitigate robust overfitting by re-striking a balance between the model trainer and the attacker. Experiments show that our proposed BoAT largely alleviates robust overfitting and attains state-of-the-art robustness.

## 2 A New Dynamic Framework for Robust and Non-robust Features

In this section, we explain how traditional static framework of feature robustness analysis fails to explain the robust overfitting problem in AT. Motivated by this fact, we establish a dynamic view of feature robustness that centers on the adversarial learning process.

### 2.1 Limitations of Traditional Static Feature Robustness Framework

The arguably most prevailing understanding of adversarial training is the feature robustness framework proposed by Ilyas *et al.* (Ilyas et al., 2019). They regard natural images as a composition of both robust and non-robust features that can both generalize (*i.e.,* useful), and the use of non-robust features in standard training results in its adversarial vulnerability. Adversarial training is to prune the non-robust features and the learned robust model only makes use of robust features. Generally, for a distribution $\mathcal{P}$ and model class $\mathcal{F}$, we can define the robustness of a feature $f \in \mathcal{F} : \mathbb{X} \to \mathbb{Y}$ as a real value, and decide whether it is a (non-)robust feature based on a given threshold $\gamma \in [0,1]$:

$$\text{(Ilyas } et \text{ } al.\text{'s definition)} \quad R_{\Delta,\mathcal{P}}(f) = \mathbb{E}_{(x,y)\sim\mathcal{P}}\left[\min_{\delta\in\Delta(x)} \mathbf{1}[f(x+\delta) = y]\right]. \tag{2}$$

**Limitations.** However, we notice that this feature robustness definition fails to explain robust overfitting. Because the robustness of a feature $f$ only depends on the attacker $\mathcal{A}$ and the distribution $\mathcal{P}$ that are both static across training, the robustness of $f$ should also be *unchanged* under different decays. The core issue here is that they only consider the attacker $\mathcal{A}$, while adversarial training is essentially a dynamic min-max game between two players, the attacker $\mathcal{A}$ (inner-loop) and the model trainer $\mathcal{T}$ (the outer-loop), both of which are typically based on first-order optimization algorithms. As a result, feature robustness will dynamically vary depending on the relative strength between $\mathcal{A}$ and $\mathcal{T}$. Another failure of Ilyas et al. (2019) is that they only consider the population-level robustness over the distribution $\mathcal{P}$ (Eq. 2) and regard robustness as an intrinsic data property. However, the large robust generalization gap in Figure 1a and 1b shows that a model can behave quite differently even on the same distribution. Therefore, we should distinguish the robustness on the training and test dataset.

To summarize, in order to explain the robust overfitting phenomenon, we need to go beyond Ilyas *et al.*'s original framework, and incorporate the adversarial game between $\mathcal{A}$ and $\mathcal{T}$ on a specific dataset $\mathcal{D}$ into the characterization of robust features, as we will do in the next part.

## 2.2 THE PROPOSED DYNAMIC FEATURE ROBUSTNESS FRAMEWORK

Different from Ilyas et al. (2019), we define a feature $g$ as a *filter* of the input $g : \mathbb{X} \to \mathbb{X}$. Canonical data features (such as, shape, texture) naturally fall into this definition. More importantly, our definition of feature is classifier-independent, which enables us to explicitly model the attacker $\mathcal{A}$ and the trainer $\mathcal{T}$ as two independent factors that affect feature robustness. Specifically, we define the robustness of feature $g$ as a function of the attack region $\Delta$, the model trainer $\mathcal{T}$, and the dataset $\mathcal{D}$:

$$(\text{our definition}) \ \mathcal{R}_{\Delta,\mathcal{T},\mathcal{D}}(g) = \max_{f_g \in \mathcal{F}_g^{\mathcal{T}}} \mathbb{E}_{x,y \in \mathcal{D}} \left[ \min_{\delta \in \Delta} \mathbf{1}[f_g(x + \delta) = y] \right], \quad (3)$$

where $\mathcal{F}^{\mathcal{T}} \subset \mathcal{F}$ denotes the attainable subset of models under the trainer $\mathcal{T}$ (a specific training configuration), and $\mathcal{F}_g^{\mathcal{T}}$ further denotes the subset of $\mathcal{F}^{\mathcal{T}}$ that utilzes feature $g$.[1] Many choices of the trainer $\mathcal{T}$ affect have a large impact on $\mathcal{F}_t$, such as the optimizer, learning rate and its schedule, weight decay, *etc*. In particular, as shown in Figure 1b, the change of learning rate has a huge impact on model robustness. Theoretically, Li et al. (2019) prove that different learning rates indeed have a strong regularization effect on NN training. Therefore, in this work, we mainly study the effect of LR decay on the change of feature robustness.

Besides the introduction of dataset $\mathcal{D}$, a key difference from Ilyas et al. (2019) is that in our framework, the robustness of a feature is not decided as *a priori*, but dependent on a dynamic game between $\mathcal{A}$ and $\mathcal{T}$. Specifically, a feature $g$ is robust if the trainer $\mathcal{T}$ can find a model $f_g$ that is resistant against the attacker $\mathcal{A}$, otherwise, $g$ is non-robust and discarded by the trainer. More importantly, this situation could change over time. When we adopt a smaller LR after decay, the trainer poses a different regularization effect on the model class $\mathcal{F}^{\mathcal{T}}$. As pointed out by Li et al. (2019), small LR often favors easy-to-generalize and hard-to-fit patterns. In the context of adversarial training, these features are often non-robust features (before LR decay). Indeed, non-robust features enjoy good generalization (Ilyas et al., 2019), but they are hard to fit under the attacker. However, with a smaller LR, the trainer becomes more capable of memorizing these features by drawing more complex decision boundaries in a small local region. Therefore, due to the change of relative strength between $\mathcal{A}$ and $\mathcal{T}$, these non-robust features could become robust after LR decay.

To verify the change of robustness, following a similar procedure as Ilyas et al. (2019) (details in Appendix B.3), we extract non-robust features from models before LR decay, and examine their robustness after LR decay . We also incorporate a control group that adopts the same training configuration except for the LR decay. Results are shown in Figure 1c. Compared with the control group with constant LR (blue line), we can see that LR decay (red line) introduces a significant increase in feature robustness on the training set ($48\% \to 77\%$). Therefore, indeed there are many non-robust features that become robust on the training set due to LR decay, as hypothesized above. However, we also notice that on the test set, robustness only changes slightly ($25\% \to 30\%$), and the non-robust features largely remain non-robust. This indicates that the robustness obtained by these non-robust features after decay are not generalizable from the training set $\mathcal{D}_{\text{train}}$ to the test set $\mathcal{D}_{\text{test}}$. The discrepancy in feature robustness shows that the rise of training robustness does not consistently

---

[1]Formally, $\forall f_g \in \mathcal{F}_g^t, \exists h : \mathbb{X} \to \mathbb{Y}$, such that $f_g = h \circ g$. Accordingly, given a threshold $\gamma$, we call a feature $g$ to be $(\Delta, \mathcal{F}, \mathcal{D})$-robust if $\mathcal{R}_{\Delta,\mathcal{F},\mathcal{D}}(g) \geq \gamma$, and a non-robust feature otherwise. We denote the set of all $(\Delta, \mathcal{F}, \mathcal{D})$-robust features as $\mathcal{G}(\Delta, \mathcal{F}, \mathcal{D})$.

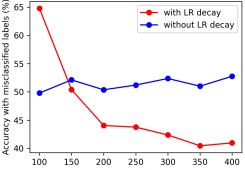 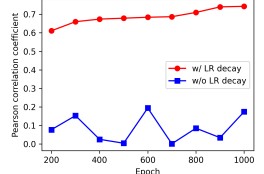 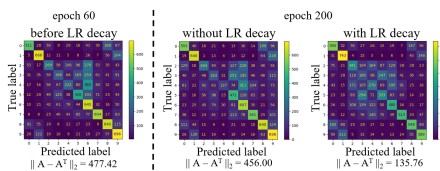

(a) target-class information in test robust features

(b) bilateral class correlation

(c) confusion matrix symmetry

Figure 2: Empirical verification of our explanation for robust overfitting. (a) After LR decays, test non-robust features become less and less informative (from 64.79% to 40.87%) of the classes to which they are misclassified. (b) Increasingly strong correlation between training-time $A \rightarrow B$ misclassification and test-time $B \rightarrow A$ misclassification increase. (c) Test-time confusion matrix of adversarial examples indeed becomes symmetric with LR decay. More details in Section 2.3.

lead to higher test robustness, leading to a large robust generalization gap. Nevertheless, we still do not fully understand why test robustness even *degrades* shortly after LR decay (Figure 1b). Below, we show that it is also induced by fitting these non-robust features with smaller LR.

## 2.3 ROBUST OVERFITTING FROM THE LENS OF DYNAMIC FEATURE ROBUSTNESS

In this section, we study how the change of feature robustness during LR decay induces the robust overfitting phenomenon. As discussed above, a significant change during LR decay is that many non-robust features become robust on the training set, while remaining non-robust on the test set. To begin with, we design a controlled experiment showing that fitting these non-robust features on the training set is strongly correlated with the degradation of robustness on the test set.

**Memorizing More Non-robust Features Brings More Harm to Test Robustness.** To study the effect of non-robust features alone, we deliberately add non-robust features (generated with different perturbation budgets $\varepsilon$) to the clean CIFAR-10 dataset, where a larger $\varepsilon$ indicates stronger non-robust features in the synthetic dataset (details in Appendix B.3). As shown in Figure 1d, with LR decays, stronger non-robust features induce severe test robustness degradation, while it is much less severe without LR decay. This shows that the memorization of non-robust features induced by LR decays indeed hurts test robustness a lot. But how does this robust overfitting phenomenon happen?

**A Hypothetical Explanation for Robust Overfitting.** From the perspective of dynamic feature robustness, the essential role of LR decay is to break the original balance between the attacker $\mathcal{A}$ and the trainer $\mathcal{T}$. The decay tips the scale towards the trainer who seizes more and more adversarial examples (containing non-robust features) and push up the training robustness. However, it is not at no cost. In the analysis below, we show that it also sets a trap for itself in the test-time attack. **1) Model Learns False Mapping of Non-robust Training Features after Decay.** Consider an adversarial example $x$ from class $A$ that is misclassified to class $B \neq A$ before decay. According to Ilyas et al. (2019), the non-robust feature $g$ of $x$ belongs to class $B$. After LR decay, the trainer $\mathcal{T}$ becomes more capable of memorization with a smaller LR, and we assume that it successfully learns to map this adversarial example $x$ to its correct label $A$. Accordingly, the non-robust feature $g$ from class $B$ in $x$ is also (falsely) mapped to class $A$. **2) False Non-robust Mapping Opens Shortcuts for Test-time Adversarial Attack.** Considering a test sample $\hat{x}$ from class $B$, before LR decay, the attacker $\mathcal{A}$ has to add non-robust features *from a different class*, say $A$, in order to misclassify it. However, after decay, because of the existence of false non-robust mapping, the attacker $\mathcal{A}$ can simply add a non-robust feature *from the same class*, denoted as $g$, such that it is misclassified to a wrong class $A$. In other words, the injected non-robust features do not even have to come from a different class, which makes it much easier for the attacker to find $g$ when starting from $\hat{x}$ (compared to non-robust features from other classes). Therefore, the attacker will have a larger attack success rate with the help of these false non-robust mappings, as they create easy-to-find shortcuts for adversarial attack. **Summary:** when the balance breaks, the trainer has an (overly) strong fitting ability such that it learns false mapping of non-robust features. At test time, these non-robust features induce shortcuts for adversarial attack, which leads to robustness degradation. From a causal perspective, these non-robust features are spurious features (Zhang et al., 2022; Sagawa et al., 2020). Therefore, when they are falsely memorized, it would also induce significant performance degradation.

**Empirical Verification.** On real-world data like images, it is hard to know what the data distribution or the non-robust features actually are. Instead, we validate this theory by verifying some nontrivial yet testable implications of it. We refer the readers to Appendix B.3 for details of these experiments.

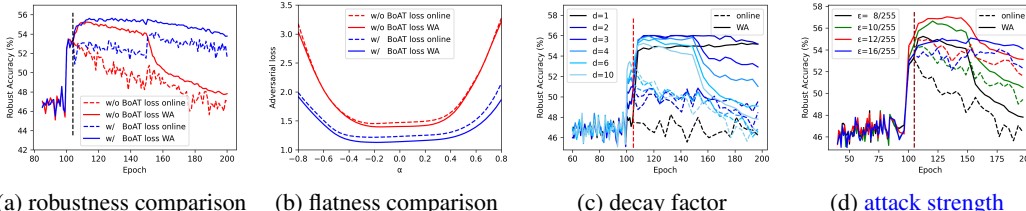

(a) robustness comparison    (b) flatness comparison    (c) decay factor    (d) attack strength

Figure 3: Mitigating robust overfitting from several different perspectives. (a, b) The purposed BoAT loss successfully boosts model robustness and mitigates robust overfitting by bootstrapping adversarial loss landscape flatness between online and WA model. (c) Small LR decay factor leads to better best-epoch robustness of WA model with only slight overfitting. (d) Stronger training attacker also helps mitigate robust overfitting and boost robustness.

**1) Lack of Target-class Features in Adversarial Examples.** In our example above, the added non-robust feature $g$ comes from the same class $B$ as $\hat{x}$, so the resulting adversarial example $\hat{x}'$ actually has little features of class $A$, even if it is (mis)classified to class $A$. Figure 2a shows that adversarial examples generated by models after LR decay indeed have less and less target-class features after the decay. **2) A Strong Bilateral Class Correlation.** Our theory suggests that if overfitting happens, the training misclassification from class $A \rightarrow B$ (before decay) will induce an increase in test misclassification from class $B \rightarrow A$ (after decay) as $A \rightarrow B$ false non-robust mappings create $B \rightarrow A$ shortcuts. We verify this in Figure 2b, where we observe that this bilateral correlation indeed consistently increases after LR decay, while remaining very low (nearly no correlation) without decay. **3) Symmetrized Confusion Matrix.** As a result of the bilateral correlation, interestingly, we can deduce that the $A \rightarrow B$ and $B \rightarrow A$ misclassification will have more similar error rates, which means that the confusion matrix of test robustness will become more symmetric. Indeed, as shown in Figure 2c, the confusion matrix (denoted as $C$) becomes much more symmetric (smaller distance between $A$ and $A^\top$) after LR decay, particularly when compared to constant learning rate.

The three experiments above help justify our understanding of robust overfitting from the perspective of dynamic feature robustness. In the next section, we further show that this theory also sheds new lights on the design of adversarial training algorithms for alleviating robust overfitting.

## 3 Mitigating Robust Overfitting from a Dynamic Perspective

In Section 2, we have introduced our dynamic feature robustness and explain robust overfitting from the shortcut effects of memorizing non-robust features. In this section, we further investigate how to alleviate robust overfitting. From the perspective of our theory, the main goal is to prevent the model trainer $\mathcal{T}$ from fitting non-robust features too quickly and too adequately. As AT is a dynamic game between the trainer $\mathcal{T}$ and the attacker $\mathcal{A}$, we can either weaken the trainer's fitting ability (Sections 3.1 & 3.2), or strengthen the attacker's attacking ability (Section 3.3), as we show below.

### 3.1 Model Regularization with Bootstrapped Flatness

In order to suppress robust overfitting of the model trainer $\mathcal{T}$ after LR decays, we need to put further regularization on its local fitting ability, *e.g.,* by enforcing better landscape flatness. In this way, we could prevent the learner from drawing complex decision boundaries and overfit to non-robust features. Indeed, Stutz et al. (2021) find that models with less robust overfitting often enjoy better landscape flatness. Among existing works, there are mainly two approaches for better flatness. One is AWP (Wu et al., 2020) by adversarially perturbing weights, and one is weight average (WA) (Izmailov et al., 2018) with equal or moving average of model parameters. Compared with AWP that introduces extra training steps and computation cost, WA is rather simple and efficient, as model average brings neglectable training overhead. Therefore, we focus on WA in this work.

Along the online model $f_\theta$ that is adversarially trained, we also maintain a WA model $f_\varphi$ that is an exponential moving average of the online parameters $\varphi \leftarrow \gamma \cdot \varphi + (1 - \gamma) \cdot \theta$, where $\gamma \in [0, 1]$ is the decay rate. Although WA can produce a flatter landscape (Izmailov et al., 2018), we notice that it is *not* enough to prevent robust overfitting. As shown in Figure 3a, the WA model still overfits shortly after the online model overfits. Rebuffi et al. (2021) further combine WA with CutMix (Yun et al., 2019), but it still cannot alleviate robust overfitting under longer training (see Figure 4a). Here, we consider an alternative solution with a new regularization loss.

We notice that when the online model $f_\theta$ deteriorates, the WA model, although flatter, will eventually overfit as it is only a moving average of $f_\theta$. Therefore, we also need to improve the flatness of the online model as well. How to achieve this? Inspired by the latent bootstrap mechanism in self-supervised learning (Grill et al., 2020), we realize that the WA model $f_\varphi$ is a flatter version of the online model $f_\theta$ itself, and thus we can bootstrap its own flatness by aligning the online model with the WA model (as a target). Specifically, given an adversarial example $x$ generated by PGD attack as in AT, we devise a new regularization term that aligns their predictions using KL divergence,

$$\ell_{\text{BoAT}}(x, y; \theta) = \ell_{\text{CE}}(f_\theta(x), y) + \lambda \cdot \text{KL}\left(f_\theta(x) \| f_\varphi(x)\right), \quad (4)$$

where $\lambda$ is a coefficient balancing the CE loss (for better training robustness) and the regularization term (for better landscape flatness). We name it as Bootstrapped Adversarial Training (BoAT) loss. Because this loss is designed to alleviate robust overfitting after decay, we only apply WA and BoAT loss after decay to save computation. We also detach the gradients from the WA model as it only serves a flatter target. Compared with CE loss, the KL regularization provides more fine-grained supervision from a flatter model that helps avoid overfitting to training labels. As a result, the online model trained with the BoAT loss will have better flatness than the vanilla AT loss. A flatter online model will further improve the flatness of the WA model, which, in return, also benefits the flatness of the online model. In this way, we could improve the flatness of both online and WA models in a bootstrapped fashion, which, consequently, alleviates the robust overfitting problems of both models.

To verify the intuition above, we compare the AT loss and the BoAT loss under the same training configurations. From Figure 3a, we can see that contrary to vanilla AT that quickly degrades after decay, the online and WA robustness of the BoAT loss continues to gradually increase after decay. In particular, with AT loss, a typical behavior is that the online model will dramatically jump to high robustness after decay with a stronger fitting ability, but because it overfits to non-robust features, its robustness also dramatically degrades shortly afterwards. Instead, with the bootstrap regularization, BoAT's online model has a slow but steady improvement of robustness, which also contributes to better robustness of the WA model. At last, BoAT's WA model achieves better robustness than AT's best WA model, showing that alleviating robust overfitting is indeed helpful for attaining better robustness. When further comparing their flatness at the last checkpoints (Figure 3b), we notice that both BoAT's online and WA models enjoy better flatness than AT's online and WA models, showing that our bootstrap mechanism effectively improves the flatness of both online and WA models.

## 3.2 SMALLER LEARNING RATE DECAY

The discussion above also suggests that a large LR decay can be harmful for robustness, as it induces severer robust overfitting. A canonical recipe of AT (Madry et al., 2018) is to adopt a piecewise linear LR schedule with a large decay (*e.g.,* LR divided by $d = 10$) at a certain epoch. As shown in Figure 3a, the online robustness quickly jumps up by nearly 10%, and Rice et al. (2020) propose to early stop the training before it deteriorates. However, our discussion on the BoAT loss shows that a longer training without overfitting might bring further improvement over this jump-and-early-stop strategy.

Apart from the regularization loss (Section 3.1), here we investigate the effect of different decay factors $d$ on robust overfitting. In our dynamic robustness framework, the LR decay plays an important role, as it breaks the balance between the trainer and the attacker by endowing the trainer with stronger local fitting ability (Section 2.2). However, we also find when it is relatively too strong, the trainer could memorize many non-robust features that are poisonous for its test robustness (Section 2.3). Therefore, a smaller decay rate with milder change of relative strength could help prevent robust overfitting. To verify this, we compare six decay factors $d$: $1, 2, 3, 4, 6, 10$, where $d = 10$ corresponds to canonical decay and $d = 1$ corresponds to constant LR (no decay). As shown in Figure 3c, the online model using smaller decay indeed has less robust overfitting, though their best robustness is also lower than $d = 10$. However, interestingly, when combined with WA, the flatter WA model can benefit a lot from smaller decay and improve both best-epoch and final-epoch robustness. Specifically, WA with a very small decay rate $d = 2$ attains better best-epoch robustness than $d = 10$ with only slight overfitting. Intuitively, because the WA model is a moving average of many online models, for better robustness and flatness, it would prefer a group of similar and growing (at least less overfitting) online models (under smaller decay) instead of a group of quickly overfitting models (under large decay). This investigation shows that with flatter landscape like WA, a smaller decay is indeed beneficial for both attaining better robustness and alleviating robust overfitting.

## 3.3 COUNTERING STRONGER MODEL TRAINER WITH STRONGER TRAINING ATTACKER

In the discussions above, we have shown that regularizing the trainer's local fitting ability with the BoAT loss or smaller learning rate decay can both help prevent it from overfitting to non-

Table 1: Comparing our method with several previous methods on CIFAR-10 under the perturbation norm $\varepsilon_\infty = 8/255$ based on PreActResNet-18 and WideResNet-34-10 architecture.

| Method | PreActResNet-18 | | | | | | WideResNet-34-10 | | | | | |
| | Natural | | PGD-20 | | AutoAttack | | Natural | | PGD-20 | | AutoAttack | |
| | best | final | best | final | best | final | best | final | best | final | best | final |
| PGD-AT | 82.08 | 83.98 | 52.64 | 47.06 | 47.72 | 42.60 | 86.09 | 86.81 | 56.45 | 48.16 | 52.11 | 45.47 |
| TRADES | 80.72 | 82.61 | 52.66 | 49.75 | 48.37 | 46.94 | 84.73 | 84.62 | 56.50 | 47.28 | 53.25 | 45.29 |
| WA | 83.37 | **85.07** | 55.24 | 47.79 | 49.92 | 43.82 | **87.84** | 86.83 | 56.21 | 49.18 | 52.41 | 46.36 |
| KD+SWA | **83.82** | 84.43 | 54.59 | 54.42 | 49.87 | 49.74 | 86.85 | **88.03** | 56.92 | 55.74 | 53.64 | 53.23 |
| PGD-AT+TE | 82.15 | 82.59 | 55.03 | 53.79 | 50.11 | 49.14 | 86.20 | 85.63 | 56.89 | 53.49 | 53.35 | 50.81 |
| AWP | 81.25 | 81.56 | 55.53 | 54.83 | 50.34 | 49.64 | 86.28 | 86.28 | 58.85 | 58.76 | 53.26 | 53.27 |
| **BoAT** | 82.20 | 82.06 | **56.59** | **56.55** | **51.39** | **51.36** | 85.21 | 85.41 | **59.82** | **59.30** | **54.94** | **54.69** |
| WA+CutMix | 80.18 | 80.18 | 56.15 | 56.11 | 49.87 | 49.87 | 85.41 | 88.30 | 60.50 | 59.32 | 55.16 | 54.10 |
| **BoAT+CutMix** | 78.75 | 78.75 | 56.30 | 56.27 | 50.41 | 50.39 | 86.49 | 86.86 | 61.72 | 61.56 | 56.13 | 56.23 |

Table 2: Comparing our method with several training methods on CIFAR-100 and Tiny-ImageNet under the perturbation norm $\varepsilon_\infty = 8/255$ based on the PreActResNet-18 architecture.

| Method | CIFAR-100 | | | | | | Tiny-ImageNet | | | | | |
| | Natural | | PGD-20 | | AutoAttack | | Natural | | PGD-20 | | AutoAttack | |
| | best | final | best | final | best | final | best | final | best | final | best | final |
| PGD-AT | 55.52 | 57.35 | 29.58 | 23.02 | 24.53 | 20.21 | 45.54 | 48.73 | 22.00 | 18.08 | 17.30 | 14.68 |
| TRADES | 55.53 | 57.09 | 29.56 | 26.08 | 24.51 | 22.86 | 47.99 | 47.79 | 22.51 | 21.28 | 16.66 | 16.29 |
| WA | **57.96** | **58.81** | 30.90 | 24.04 | 25.95 | 21.02 | 48.76 | 50.00 | 24.66 | 19.82 | 19.76 | 15.82 |
| KD+SWA | 57.23 | 57.66 | 30.06 | 30.02 | 26.04 | 25.99 | **50.59** | 50.83 | 24.83 | **24.79** | 19.78 | 19.76 |
| PGD-AT+TE | 56.52 | 57.30 | 31.23 | 29.25 | 26.04 | 25.13 | 46.84 | 50.51 | 22.89 | 19.40 | 18.16 | 15.88 |
| AWP | 53.92 | 54.81 | 30.67 | 30.43 | 25.26 | 25.07 | 43.31 | 43.30 | 23.60 | 23.26 | 18.08 | 17.84 |
| **BoAT** | 56.44 | 56.51 | **32.70** | **32.38** | **27.59** | **27.52** | 47.79 | 48.20 | **25.21** | 24.52 | **20.41** | **19.88** |
| WA+CutMix | 56.84 | 56.96 | 32.35 | 32.34 | 26.39 | 26.35 | 46.58 | 46.87 | 24.97 | 24.87 | 18.97 | 18.87 |
| **BoAT+CutMix** | 55.74 | 55.88 | 33.03 | 33.02 | 27.21 | 27.20 | 46.02 | 46.15 | 25.04 | 25.02 | 19.51 | 19.50 |

robust features. As noted in our dynamic framework, AT is a dynamic game between the trainer $\mathcal{T}$ and the attacker $\mathcal{A}$. Another natural approach is to strengthen the attacker's attacking ability to counter the stronger trainer after decay, which would also help strike a new balance between two players. To examine this, we apply four different training perturbation budgets after decay, $\varepsilon = 8/255, 10/255, 12/255, 16/255$ (with fixed step size $\alpha = 2/255$ and increasing PGD steps $k = 10 \cdot \varepsilon/(8/255)$), where $\varepsilon = 8/255$ is the default budget used before decay and in test-time attack. As can be seen from Figure 3d, models trained with stronger attacker (larger $\varepsilon$) indeed has much less overfitting, which helps verify our understanding of robust overfitting through the dynamic game perspective. Besides, a too strong training attacker also prevents the trainer from learning useful robust features (under $\varepsilon = 16/255$). Empirically, we observe that selecting a moderately larger attack strength, $\varepsilon = 12/255$, attains the optimal best-epoch robustness (see also Appendix B.4). This shows that we need to re-design the training attacker according to the change of the model trainer.

### 3.4 OVERALL APPROACH

Above, we have introduced three effective techniques to mitigate robust overfitting from different perspectives: model regularization, learning rate decay, and attacker strength. At last, we combine the BoAT loss and smaller learning rate as our final approach. We do not include stronger attacker because stronger attacker often induces a large decrease in natural accuracy. This is because stronger attacks will make all features less robust, which will decrease the number of usable features in the classifier and hurt natural accuracy, although it could further bring a slight increase of robustness. We include a study on this trade-off in Appendix B.4.

## 4 EXPERIMENTS ON BENCHMARK DATASETS

In this section, we evaluate the effectiveness of the proposed BoAT on several benchmark datasets using different network architectures.

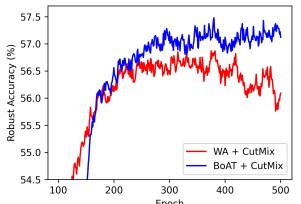
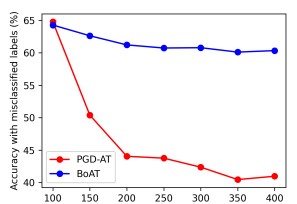
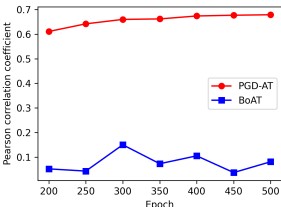

(a) training longer with CutMix    (b) target-class information in test    (c) bilateral class correlation
non-robust features

Figure 4: Empirical understandings on BoAT. (a) BoAT can benefit from training longer (with CutMix). (b) Compared to PGD-AT, test adversarial examples still have to contain fairly much target-class information (measured by accuracy) under BOAT. (c) BoAT does not result in strong bilateral class correlation as in PGD-AT.

**Setup.** We consider the classification tasks on CIFAR-10, CIFAR-100 (Krizhevsky et al., 2009), and Tiny-ImageNet (Deng et al., 2009) with the PreActResNet-18 (He et al., 2016) and WideResNet-34-10 (Zagoruyko & Komodakis, 2016) architectures. Besides the vanilla PGD-AT (Madry et al., 2018), we include the following baseline methods for alleviating robust overfitting: WA (Izmailov et al., 2018), KD+SWA (Chen et al., 2021), PGD-AT+TE (Dong et al., 2022), AWP (Wu et al., 2020) and WA+CutMix (Rebuffi et al., 2021). During evaluation, we use PGD-20 (Madry et al., 2018) and AutoAttack (AA) (Croce & Hein, 2020) for the adversarial attack methods. We train each model for 200 epochs and record the *best-epoch* (selected based on PGD attack) and *final-epoch* robust test accuracy. Please refer to Appendix B.5 and B.6 for details and Appendix C for additional results.

## 4.1 RESULTS ON BENCHMARK DATASETS

**Performance across Different Networks.** In Table 1, we evaluate BoAT against previous AT variants on two popular backbone networks: PreActResNet-18 and WideResNet-34-10. We can see that without strong data augmentations like CutMix, BoAT attains state-of-the-art robustness on both models and outperforms previous methods by *at least 1%* against AutoAttack (AA), arguably the strongest white-box attacker. Meanwhile, BoAT almost alleviates robust overfitting on PreActResNet-18 with only 0.03% gap (51.39% best and 51.36% final), and the gap is also small (0.30%) on WideResNet. Because of this, remarkably, the final-epoch robustness of BoAT even consistently outperforms the best-epoch robustness of previous approaches using early stopping. Therefore, with BoAT, we might not need per-epoch validation and early stopping techniques any more, which helps save training computation cost, simplify the training recipe, and improve the applicability of AT.

**Strong Augmentations.** WideResNet is more prone to overfit as it has larger capacity, and existing WA+CutMix strategy (Rebuffi et al., 2021) still overfits by more than 1% (55.16% best and 54.10% last). In comparison, BoAT+Cutmix achieves 56.13% best robustness (chosen according to PGD) and 56.23% final robustness, showing that this combination indeed alleviates robust overfitting on WideResNet and attain state-of-the-art robustness. In comparison, CutMix brings little improvement on PreActResNet-18, mainly because this model has lower capacity and still underfits under strong augmentations, and it requires longer training to reach higher performance. Even though, BoAT+CutMix still outperforms WA+CutMix by 0.5% AA robustness by the end of 200 epochs.

**Performance across Different Datasets.** We also conduct experiments on two additional benchmark datasets, CIFAR-100 and Tiny-ImageNet, that are more complex than CIFAR-10. As shown in Table 2, our BoAT and BoAT+Cutmix still achieve the highest best and last robustness, demonstrating its scalability to larger scale datasets. Similarly, CutMix does not help much as it often leads to underfitting with stronger regularization effects within a relatively short training length.

## 4.2 EMPIRICAL UNDERSTANDINGS

**BoAT Can Benefit from Longer Training.** Table 1 suggests that when CutMix is applied to a PreActResNet-18 model, it demands longer training to fully unleash the beast. This motivates us to examine the effect of longer training, *e.g.,* 500 epochs. As shown in Figure 4a, longer training with WA+CutMix brings little improvement on final robustness (49.87% (200 epochs) *v.s.* 49.90% (500 epochs) under AA). Instead, BoAT+CutMix achieves higher best robustness and the robustness keeps improving along training (> 1% improvement under AA). In particular, the additional 300 epochs improve the final AA robustness of BoAT+CutMix from 50.39% to 51.28%. This provides strong

evidence on the effectiveness of BoAT against robust overfitting. It also shows that combining BoAT with strong data augmentations can benefit from long training as in standard training.

**Verification on Robust Overfitting Experiments.** Beyond its success, we empirically study how BoAT manages to suppress robust overfitting through a lens of feature robustness. To this end, following the experiments in Section 2.3, we compare PGD-AT and BoAT in terms of 1) the amount of target-class features of test adversarial examples and 2) the strength of bilateral class correlation. First, we can see that test adversarial examples still have to contain much target-class features to be misclassified (Figure 4b) and there is no longer a strong bilateral class correlation (Figure 4c). This confirms that BoAT indeed successfully prevent the model from learning false mapping of non-robust training features which creates shortcuts for test-time attack.

## 5 REVISITING EXISTING APPROACHES TO ALLEVIATE ROBUST OVERFITTING

Remind that our definition of feature robustness (Eq. 3) is composed of three factors: the dataset $\mathcal{D}$, the model trainer $\mathcal{T}$, and attacker $\mathcal{A}$. Based on our understanding of robust overfitting (RO), we can categorize existing AT methods that help alleviate RO into the following three classes.

**Data Regularization.** As RO is induced by non-robust features in training *inputs*, one way is to corrupt non-robust features with various input transformations. As can be seen from Ilyas et al. (2019), non-robust features often correspond to local and high-frequency patterns in images. Therefore, stronger augmentations like CutMix will help mitigate RO, as analyzed by Rice et al. (2020). Recently, Li et al. (2022) show that random or even fixed masks can eliminate the catastrophic overfitting of FGSM-AT. As the FGSM attacker is much weaker than PGD, FGSM-AT would be more prone to overfit, and we reckon that the masks help corrupt non-robust features in original data. A complementary approach is to correct the *supervision* of the adversarial examples accordingly to avoid learning false label mapping (Section 2.3), *e.g.,* the soft labels adopted in Dong et al. (2021), and the distillation-alike methods in KD+SWA (Chen et al., 2021) and TE (Dong et al., 2022).

**Training Regularization.** A necessary condition of RO is that the model trainer is stronger than the attacker, which breaks their original balance. Therefore, we can prevent RO by regularizing the trainer to be relatively weaker on fitting non-robust features. One approach is to directly promote model flatness, *e.g.,* AWP (Wu et al., 2020), EMA Rebuffi et al. (2021), SWA Chen et al. (2021), and our BoAT loss (Section 3.1). Yu et al. (2022) propose to regularize samples of small loss values (typically using non-robust features). Another approach is to adjust the learning rate schedule to avoid dramatic change of robustness that could be irreparable, as studied in Section 3.2. Previously, Rice et al. (2020) and Wang & Wang (2022) also study different types of learning schedules, while we focus on the effect of LR decay factors on robust overfitting. In particular, we firstly show that piecewise decay with a small decay factor can be helpful for both better robustness and less RO.

**Stronger Training Attacker.** Another way to restore the balance is to adopt a *stronger* training attacker *after LR decay*, as we have explored in Section 3.3. Previously, from a curriculum training perspective, Cai et al. (2018), Qin et al. (2019), and Wang et al. (2019) propose to start training from a *weaker* attacker (*e.g.,* $\varepsilon = 0$), and gradually lift its strength to the standard level (*e.g.,* $\varepsilon = 8/255$). These strategies are designed as a warmup stage at the earlier training stages. Pang et al. (2020) point out that these warmup methods are hardly useful when measured by AA robustness. Instead, our method is designed from a dynamic game perspective, and we propose to adopt a *stronger* training attacker ($\varepsilon > 8/255$) *after the LR decay* in order to counter robust overfitting. Besides, our method shows clear benefits on improving best AA robustness as well as mitigating robust overfitting.

## 6 CONCLUSION

In this paper, we investigate the underlying cause of robust overfitting through the lens of dynamics feature robustness. To achieve this, we have proposed a new dynamic definition of feature robustness by taking the attacker $\mathcal{A}$, the model $\mathcal{M}$, and the dataset $\mathcal{D}$ into consideration. Using our new framework, we analyze the change of robust and non-robust features during the learning rate decay, and attribute robust overfitting to the falsely memorized non-robust features. Based on this understanding, we have proposed three new approaches to restore the balance between the model trainer and the attacker. Experiments on benchmark datasets show that our proposed Bootstrapped Adversarial Training (BoAT) largely resolves the overfitting issue, even under very long training. Meanwhile, it also achieves state-of-the-art robustness even without early stopping, showing that adversarial training can also benefit from longer training as long as the overfitting issue is resolved.

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

## A ADDITIONAL EXPERIMENT: ADVERSARIAL EXAMPLES (IN ADVERSARIAL TRAINING) CONTAIN NON-ROBUST FEATURES

At the end of Section 2.2, we craft adversarial examples on model $A$ and evaluate the misclassified ones on another model $B$ with their correct labels to indicate the amount of features that have become robust. We argue that this is reasonable because adversarial examples used for AT consist of robust features from the original label $y$ and non-robust features from the misclassified label $\hat{y}$, so $B$ must have robustified those features (non-robust on $A$) to correctly classify them. In this section, we further provide a rigorous discussion on this through an additional experiment.

**Experiment Design.** Recall that Ilyas et al. (2019) leverage targeted PGD attack (Madry et al., 2018) to craft adversarial examples on a standard (non-robust) classifier and relabel them with their target labels to build a non-robust dataset. Finding standard training on it yields good accuracy on clean test data, they prove that those adversarial examples contain non-robust features corresponding to the target labels (Section 3.2 of their paper).

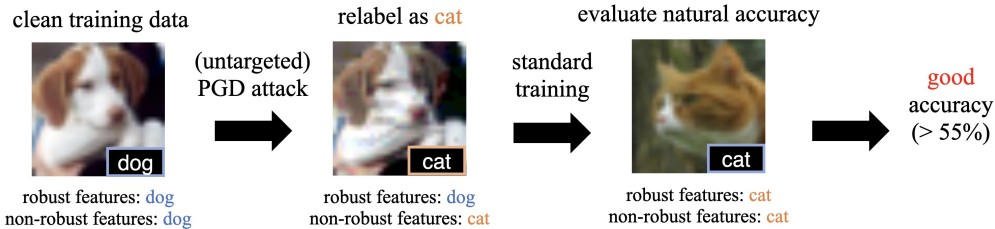

Figure 5: Mining non-robust features from adversarial examples in adversarial training. 1) Craft adversarial examples $\hat{x}_i$ using untargeted PGD attack on a AT checkpoint. 2) Relabel them (only misclassified ones) with misclassified label $\hat{y}_i$ to build a non-robust dataset $\{(\hat{x}_i, \hat{y}_i)\}$. 3) Perform standard training from the checkpoint. 4) Achieve good natural accuracy ($> 55\%$).

However, different from their settings where non-robust features lie statically in the examples, we lay emphasis on mining non-robust features from adversarial examples generated on-the-fly in AT. To this end, we first craft adversarial examples on the AT checkpoint at some epoch using untargeted PGD attack (Madry et al., 2018) to restore the real setting of AT, then relabel the *misclassified* ones $\hat{x}_i$ (*e.g.,* a dog) with their misclassified labels $\hat{y}_i \neq y_i$ (*e.g.,* cat) to build a non-robust dataset $\{(\hat{x}_i, \hat{y}_i)\}$. Also, in order to capture non-robust features at exactly the training epoch these adversarial examples are used, we continue to perform standard training directly from the checkpoint on the non-robust dataset, and finally evaluate natural accuracy. See Figure 5 for an illustration of our experiment.

In details, we select the AT models at epoch 60 (before LR decay) and epoch 1,000 (after LR decay) to conduct the above experiment. We use untargeted PGD-20 with perturbation norm $\varepsilon_{\infty} = 16/255$ to craft adversarial examples on those checkpoints from the *training data* of CIFAR-10 (Krizhevsky et al., 2009). The attack we adopt is a little bit stronger than the attack of PGD-10 with $\varepsilon_{\infty} = 8/255$ that is commonly used to generate adversarial examples in baseline adversarial training, because the training robust accuracy rises to as high as $94.67\%$ at epoch 1,000 (Figure 1a) and gives less than 2,700 misclassified examples, which is significantly insufficient for further training. Using the stronger attack, we obtain a success rate of $78.38\%$ on the checkpoint at epoch 60 and a success rate of $34.86\%$ on the checkpoint at epoch 1,000. Since the attack success rates are different, we randomly select a same number of misclassified adversarial examples for standard training for a fair comparison. The learning rate is initially set to 0.1 and decays to 0.01 after 20 epochs for another 10 epochs of fine-tuning. Given that the original checkpoints already have nontrivial natural accuracy, we also add two control groups that train with random labels instead of $\hat{y}$ to exclude the influence that may brought by the original accuracy.

**Results.** As shown in Figure 6, we find that standard training on the non-robust dataset $\{(\hat{x}_i, \hat{y}_i)\}$ successfully converges to fairly good accuracy ($> 55\%$) on natural test images, *i.e.,* predicting cats as cats, no matter from which AT checkpoints (either epoch 60 or epoch 1,000) the adversarial examples are crafted. Also, we can see that the nontrivial natural accuracy has nothing to do with the original accuracy of the AT checkpoints, as the accuracy plummets to around 10% (random guessing) as soon as the standard training starts with random labels. This proves that *adversarial examples in adversarial training do contain non-robust features w.r.t. the classes to which they are misclassified*, which strongly corroborates the validity of the experiments in Section 2.2.

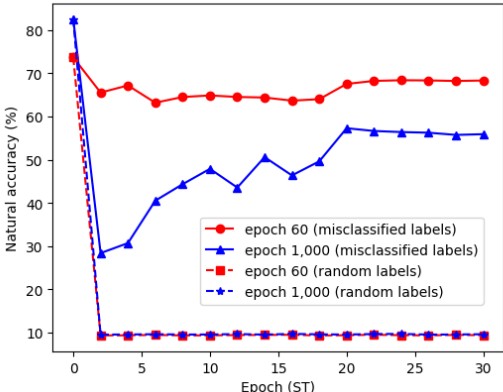

Figure 6: Natural accuracy during standard training. Standard training on the non-robust dataset built from the checkpoint at either epoch 60 or epoch 1,000 converges to fairly good natural accuracy ($> 55\%$). The failure of training with random labels proves that the good accuracy has nothing to do with the original natural accuracy of the AT checkpoint.

**A Discussion on the Influence Brought by Robust Features During Standard Training.** Since untargeted PGD attack cannot be assigned with a target label, we cannot guarantee that the misclassified labels $\hat{y}$ to be uniformly distributed regardless of the original labels (especially for real-world datasets). This implies that the non-robust features are not completely decoupled from robust features, *i.e.,* training on $\{(\hat{x}_i, \hat{y}_i)\}$ may take advantage of $\hat{x}_i$'s robust features from $y_i$ through the correlation between $y_i$ and $\hat{y}_i$. However, we argue that robust features from $y_i$ mingling with non-robust features from $\hat{y}_i$ only increases the difficulty of obtaining a good natural accuracy. This is because learning through the shortcut will only wrongly map the robust features from $y_i$ to $\hat{y}_i$ that never equals to $y_i$, but during the evaluation, each clean test example $x_i$ will always have robust and non-robust features from $y_i$, and such wrong mapping will induce $x_i$ to be misclassified to some $\hat{y}_i$ due to the label of its robust features. As a result, despite the negative influence brought by robust features, we still achieve good natural accuracy at last, which even solidifies our conclusion.

# B EXPERIMENT DETAILS

In this section, we provide more experiment details that are omitted before due to the page limit.

## B.1 BASELINE ADVERSARIAL TRAINING

In this paper, we mainly consider classification task on CIFAR-10 (Krizhevsky et al., 2009). The dataset contains 60,000 $32 \times 32$ RGB images from 10 classes. For each class, there are 5,000 images for training and 1,000 images for evaluation.

For baseline adversarial training, We use PreActResNet-18 (He et al., 2016) model as the classifier. We use PGD-10 attack (Madry et al., 2018) with step size $\alpha = 2/255$ and perturbation norm $\varepsilon_\infty = 8/255$ to craft adversarial examples on-the-fly. We use SGD optimizer with momentum 0.9, weight decay $5 \times 10^{-4}$ and batch size 128 to train the model for as many as 1,000 epochs. The learning rate (LR) is initially set to be 0.1 and decays to 0.01 at epoch 100 and further decays to 0.001 at epoch 150. For the version without LR decay used for comparison in our paper, we simply keep the LR to be 0.1 during the whole training process.

Each model included in this paper is trained on a single NVIDIA GeForce RTX 3090 GPU. For PGD-AT, it takes about 3d 14h to finish 1,000 epochs of training.

## B.2 LEARNING RATE DECAY INDUCES ROBUST OVERFITTING

In this paper, one of our most essential observations is that RO often happens after LR decays. We further verify that this is the general case instead of a phenomenon that happens to occur only when

the LR first decays at epoch 100 by allowing it to decay at different times during the training. Figure 7 demonstrates that whenever the LR decays, the test robust accuracy immediately increases by more than 5% and then slowly drops, implying that RO is indeed caused by LR decay.

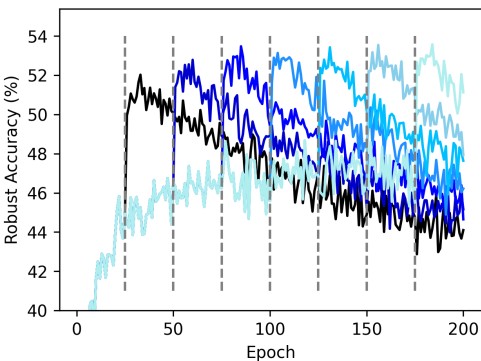

Figure 7: Test robust accuracy on CIFAR-10 under the perturbation norm $\varepsilon_\infty = 8/255$ based on the PreActResNet-18 architecture. The gray vertical lines mark the epochs where the LR decays.

### B.3 DYNAMIC FEATURE ROBUSTNESS

**Features of the Training Data Become Robust After LR Decay.** We craft adversarial examples on a checkpoint before LR decay and evaluate the *misclassified* ones on a checkpoint after LR decay with their correct labels to evaluate the amount of features that have become robust (see Appendix A for detailed discussions) at the end of Section 2.2. Following Appendix A, we adopt PGD-20 attack with perturbation norm $\varepsilon_\infty = 16/255$ to craft adversarial examples which is stronger than the common attack setting we use in PGD-AT. We note that test adversarial examples crafted by a stronger attack indicates stronger extraction of their non-robust features, so they are more indicative of feature robustification when still correctly classified.

**Memorizing Non-robust Features Harms Test Robustness.** At the beginning of Section 2.3, we create synthetic datasets to demonstrate that memorizing the non-robust training features indeed harms test-time model robustness. To instill non-robust features into the training dataset, we minimize the adversarial loss *w.r.t.* the training data in a way that just like PGD attack, with the only difference that we minimize the adversarial loss instead of maximizing it. Since we only use a very small perturbation norm $\varepsilon_\infty \leq 4/255$, the added features are bound to be non-robust. For a fair comparison, we also perturb the training set with random uniform noise of the same perturbation norm to exclude the influence brought by (slight) data distribution shifts. We continue training from the 100-th baseline AT checkpoint (before LR decay) on each synthetic dataset for 10 epochs, and then evaluate model robustness with clean test data.

**Verification: Lack of Target-class Features.** This is to say that when RO happens, we expect that test adversarial examples become less informative of the classes to which they are misclassified according to our theory. To verify this, we first craft adversarial examples on a checkpoint $T$ after RO begins, then evaluate the misclassified ones *with their misclassified labels $\hat{y}$* on the checkpoint saved at epoch 60. As a result, the accuracy reflects how much features from $\hat{y}$ the examples have to contain to be misclassified on $T$. All adversarial examples evaluated in the experiments in Section 2.3 are crafted using PGD-10 attack with perturbation norm $\varepsilon_\infty = 8/255$.

**Verification: Bilateral Class Correlation.** To quantitatively analyse the correlation strength of bilateral misclassification described in Section 2.3, empirical verification 2, we first summarize all $y_i \rightarrow y_j$ misclassification rates into two confusion matrices $P^{\mathrm{train}}, P^{\mathrm{test}} \in \mathbb{R}^{C \times C}$ for the training and test data, respectively. Because we are mainly interested in the effect of the LR decay, we focus on the relative change on the test data, *i.e.,* $\Delta P^{\mathrm{test}} = P^{\mathrm{test}}_{\mathrm{new}} - P^{\mathrm{test}}_{\mathrm{old}}$. According to our analysis above, for each class pair $(i, j)$, there should be a strong correlation between the training misclassification of $i \rightarrow j$ before LR decays, *i.e.,* $(P^{\mathrm{train}}_{\mathrm{old}})_{ij}$, and the *increase* of test misclassification of $j \rightarrow i$, *i.e.,* $\Delta P^{\mathrm{test}}_{ji}$. To examine their relationship, we plot the two variables $((P^{\mathrm{train}}_{\mathrm{old}})_{ij}, \Delta P^{\mathrm{test}}_{ji})$ and compute their Pearson correlation coefficient $\rho$.

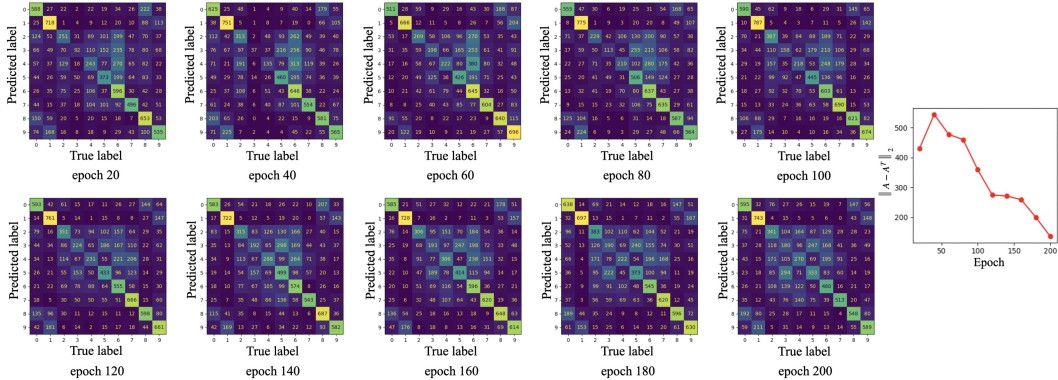

Figure 8: More test-time confusion matrices during the first 200 epochs of the training. After LR decays (the second row), the confusion matrix $A$ immediately becomes symmetric, as the spectral norm $\|A - A^T\|_2$ *w.r.t.* the matrix decreases from $\geq 350$ before epoch 100 to $\leq 150$ after epoch 200.

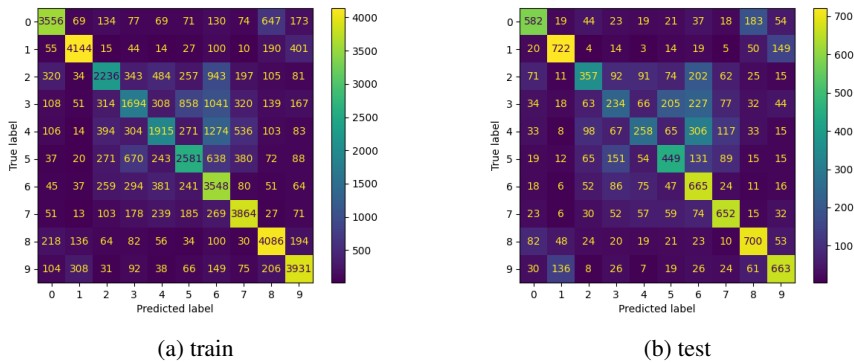

Figure 9: Confusion matrices of the training and the test data at an epoch before robust overfitting starts. They show nearly a same pattern of attacking preference among classes.

**Verification: Symmetrized Confusion Matrix.** In Section 2.3, we mention the growing symmetry of the test-time confusion matrix after LR decays as an evidence of the strengthening $A \to B$ and $B \to A$ correlation. Here we present more confusion matrices during the first 200 epochs of the training in Figure 8, and it is very clear that the confusion matrices soon become symmetric after LR decays at epoch 100 when robust overfitting starts. For a deeper comprehension of this phenomenon, we first visualize the confusion matrices of the training and the test data at an epoch before robust overfitting starts in Figure 9. They exhibit nearly a same pattern of attacking preference among classes (*e.g.*, $A \to B$) due to the bias rooted in the dataset, *e.g.*, class 6 is intrinsically vulnerable in this case. For the test data, this intrinsic bias wouldn't be wiped out through learning due to the non-generalizability of robustified training data features demonstrated in Section 2.2 (*i.e.*, $A \to B$ still holds); and for the training data, this biased feature memorization will open shortcuts for test-time adversarial attack as discussed in Section 2.3 (*i.e.*, $B \to A$ begins). Combining both $A \to B$ and $B \to A$, we arrive at the symmetry of test-time confusion matrix.

**Changing LR Decay Schedule.** Our discussion above is based on the piecewise LR decay schedule, in which the sudden decay of LR most obviously reflects our understandings. Besides, we also explore using other decay schedules, including Cosine/Linear LR decay, to check whether different decay schedules will affect the observations and claims we made in this paper. For each schedule, we train the model for 200 epochs following the settings in Rice et al. (2020). As demonstrated in Figure 10a, we arrive at the same finding as Rice et al. (2020) that with Cosine/Linear LR decay schedule, the training still suffers from severe RO after epoch 130. Then, we rerun the three empirical experiments in Section 2.3 and find that under both two LR decay schedules 1) the test adversarial examples indeed contain less and less target-class information as the training goes (Figure 10b), 2) the bilateral class correlation becomes increasingly strong (Figure 10c) and 3) the confusion matrix

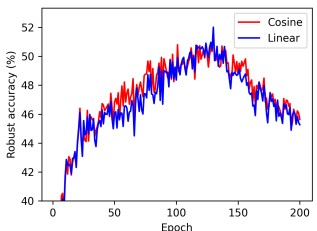 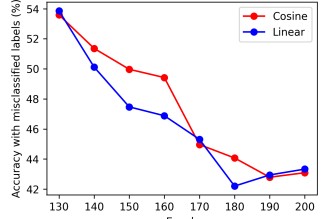 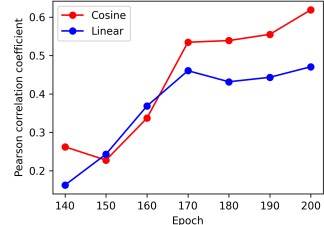

(a) test robust accuracy during training

(b) target-class information in test robust features

(c) bilateral class correlation

Figure 10: Empirical verification of our explanation for robust overfitting when Cosine/Linear LR decay schedule is applied. (a) With Cosine/Linear LR decay schedule, the training still suffers from severe RO. (b) After RO begins, non-robust features in the test data become less and less informative of the classes to which they are misclassified. (c) Increasingly strong correlation between training-time $A \rightarrow B$ misclassification and test-time $B \rightarrow A$ misclassification increase.

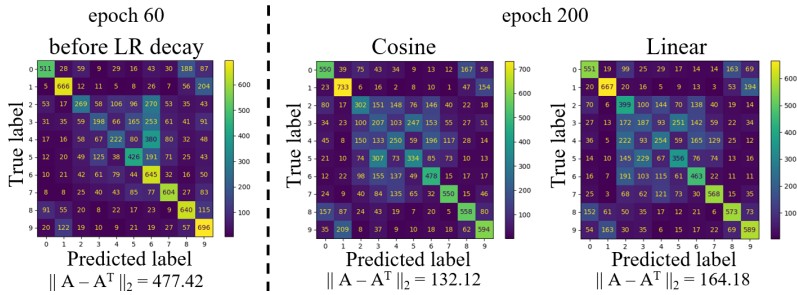

Figure 11: Test-time confusion matrix also becomes symmetric and implies that the bilateral correlation also exists when Cosine/Linear LR decay schedule is applied.

indeed becomes symmetric (Figure 11). The results are almost the same as the results achieved when we adopt piecewise LR decay schedule, indicating that our understandings in the cause of RO is fundamental and regardless of whatever LR decay schedule is used.

## B.4 The Effect of Stronger Training Attacker

**Stronger Attack for Stronger Defense.** In Section 3.3, we point out that using a stronger attacker in AT is able to mitigate RO to some extent by neutralizing the learning power when it is overly strong. To achieve the results reported in Figure 3d, we craft adversarial examples on-the-fly with more PGD iteration steps when $\varepsilon$ is larger (see Table 3), and further evaluate the best and last robustness of the WA models against PGD-20 and AA.

Table 3: Training with stronger attack and evaluating model robustness on CIFAR-10 under the perturbation norm $\varepsilon_\infty = 8/255$ based on the PreActResNet-18 architecture.

| Attack Strength | Natural | | | PGD-20 | | | AutoAttack | | |
|---|---|---|---|---|---|---|---|---|---|
| | best | final | diff | best | final | diff | best | final | diff |
| $\varepsilon = 8/255$, PGD-10 (baseline) | **83.37** | **85.07** | **-1.70** | 55.24 | 47.79 | 7.45 | 49.92 | 43.82 | 6.10 |
| $\varepsilon = 10/255$, PGD-12 | 80.66 | 82.48 | -1.82 | 56.63 | 50.63 | 6.00 | 50.89 | 46.13 | 4.76 |
| $\varepsilon = 12/255$, PGD-15 | 78.17 | 80.25 | -2.08 | **57.09** | 53.13 | 3.96 | **50.99** | 47.66 | 3.33 |
| $\varepsilon = 14/255$, PGD-17 | 73.92 | 76.70 | -2.78 | 56.42 | 54.04 | 2.38 | 50.28 | 48.53 | 1.75 |
| $\varepsilon = 16/255$, PGD-20 | 69.51 | 73.11 | -3.60 | 55.09 | **54.27** | **0.82** | 49.58 | **48.53** | **1.05** |

Although RO is only partially mitigated and natural accuracy decreases when a stronger attacker is applied, it is surprising to find from Table 3 that an attacker of appropriate strength may significantly boost the best robustness. This suggests using a stronger attack could potentially be an interesting new path to stronger adversarial defense, and we leave it for future work.

**Using Stronger Training Adversarial Attack in BoAT.** Above we show BoAT is almost capable of eliminating RO. However, since we also show leveraging stronger adversarial attack in AT may help mitigate RO while boosting model robustness above, it is natural to further combine BoAT with this training scheme. According to Figure 12, though a slightly stronger attack (*e.g.,* $\varepsilon = 9/255$) may marginally improves the best and last robust accuracy, it heavily degrades natural accuracy, particularly when much stronger attack is used. We deem that this is because it breaks the balance that BoAT has achieved: with relatively weak learning ability of the trainer $\mathcal{T}$, a stronger attacker $\mathcal{A}$ naturally dominates the adversarial game and results in an underfitting state that harms both robust and natural accuracy.

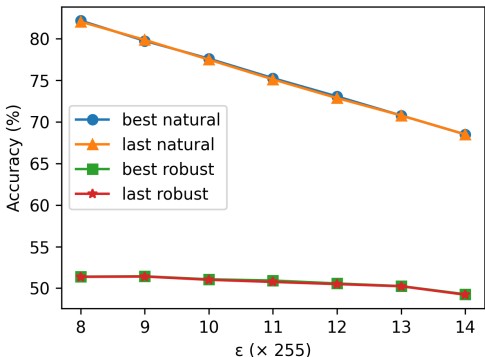

Figure 12: Using different adversarial attack strength in BoAT.

### B.5 MITIGATING ROBUST OVERFITTING WITH BoAT

**Datasets.** Beside CIFAR-10, we also include CIFAR-100 (Krizhevsky et al., 2009) and Tiny-ImageNet (Deng et al., 2009) for evaluation of the effectiveness of BoAT. CIFAR-100 shares the same training and test images with CIFAR-10, but it classifies them into 100 categories, *i.e.,* 500 training images and 100 test images for each class. Tiny-ImageNet is a subset of ImageNet (Deng et al., 2009) which contains labeled $64 \times 64$ RGB images from 200 classes. For each class, it includes 500 and 50 images for training and evaluation respectively.

**Training Strategy.** For CIFAR-10 and CIFAR-100, we follow exactly the same training strategy as introduced in B.1. For Tiny-ImageNet, we follow the learning schedule of Chen et al. (2021), in which the model is trained for a total of 100 epochs and the LR decays twice (by 0.1) at epoch 50 and 80.

**Choices of Hyperparameters.** For KD+SWA (Chen et al., 2021), PGD-AT+TE (Dong et al., 2022), AWP (Wu et al., 2020) and WA+CutMix (Rebuffi et al., 2021), we strictly follow their original settings of hyperparameters. SWA/WA as well as the BoAT regularization purposed in Section 3.1 starts at epoch 105 (5 epochs later than the first LR decay where robust overfitting often begins), and following (Rebuffi et al., 2021) we choose the EMA decay rate of WA to be $\gamma = 0.999$. Please refer to Table 4 for our choices of the decay factor $d$ and regularization strength $\lambda$. We notice that since CutMix improves the difficulty of learning, the model demands a relatively larger decay factor to better fit the augmented data. For Tiny-ImageNet, we also apply a larger $\lambda$ after the second LR decay to better maintain the flatness of adversarial loss landscape and control robust overfitting.

Table 4: Choices of hyperparameters when training models on different datasets using different network architectures with BoAT.

| Network Architecture | Method | CIFAR-10/CIFAR-100 | Tiny-ImageNet |
|---|---|---|---|
| PreActResNet-18 | BoAT | $d = 1.5, \lambda = 1.0$ | $d = 4.0, \lambda_1 = 2.0, \lambda_2 = 10.0$ |
| | BoAT+Cutmix | $d = 4.0, \lambda = 2.0$ | $d = 6.0, \lambda = 1.5$ |
| WideResNet-34-10 | BoAT | $d = 1.3, \lambda = 0.5$ | - |
| | BoAT+Cutmix | $d = 4.0, \lambda = 2.0$ | - |

### B.6 TRAINING ROBUST ACCURACY

Figure 13 shows the robust accuracy change on the training data during the training. Compared with vanilla PGD-AT that yields training robust accuracy over 80% at epoch 200, BoAT manages to suppress it to only 65%. It successfully prevents the model from learning the non-robust features *w.r.t.* the training data too fast and too well, and therefore significantly reduces the robust generalization gap (from $\sim 35\%$ to $\sim 9\%$) and mitigates robust overfitting.

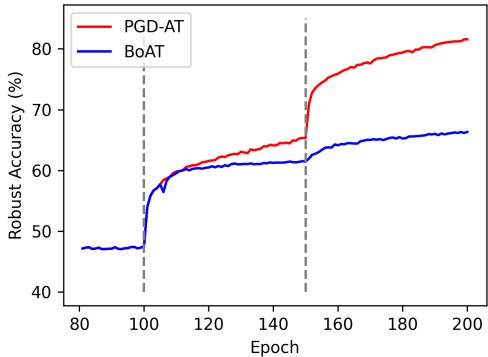

Figure 13: Training robust accuracy of PGD-AT and BoAT on CIFAR-10 under the perturbation norm $\varepsilon_\infty = 8/255$ based on the PreActResNet-18 architecture.

## C    MORE EXPERIMENTS ON BOAT

In this section, we conduct extensional experiments on the proposed BoAT method to further demonstrate its effectiveness, efficiency and flexibility.

### C.1   ABLATION STUDY

**Effectiveness of BoAT Loss.** In Section 3.1, Figure 3a has demonstrated the effectiveness of BoAT loss. Table 5 further report the detailed robustness against PGD-20 and AA, showing that the purposed regularization not only boosts the best robustness by a large margin (0.67% higher against AA) but also significantly suppresses RO (1.52% *v.s.* 6.10% against AA). To achieve the reported robustness, we first use $\lambda_1 = 10.0$ after the first LR decay and then apply $\lambda_2 = 60.0$ after the second LR decay to better maintain the flatness of adversarial loss landscape and control robust overfitting.

Table 5: Comparing model robustness w/ and w/o BoAT loss on CIFAR-10 under the perturbation norm $\varepsilon_\infty = 8/255$ based on the PreActResNet-18 architecture.

| Method | Natural | | | PGD-20 | | | AutoAttack | | |
|---|---|---|---|---|---|---|---|---|---|
| | best | final | diff | best | final | diff | best | final | diff |
| WA | **83.37** | **85.07** | -1.70 | 55.24 | 47.79 | 7.45 | 49.92 | 43.82 | 6.10 |
| BoAT($d=10$) | 81.54 | 82.42 | **-0.88** | **55.58** | **53.74** | **1.84** | **50.59** | **49.07** | **1.52** |

**Larger Decay Factor Goes with Stronger BoAT Regularization.** Below we show that when a relatively large decay factor $d$ is applied, *i.e.,* the model has overly strong fitting ability that results in robust overfitting, a large regularization coefficient $\lambda$ should be choosen for better performance. Table 6 reveals this relationship between $d$ and $\lambda$. When $d = 1.3$, even a $\lambda$ as small as 1.0 will harm both the best and last robustness as well as natural accuracy, as $d = 1.3$ is already too small a decay factor that makes the model suffering from underfitting and naturally requiring no more flatness regularization. On the other side, when $d$ is relatively large, even a strong regularization of $\lambda = 4.0$ is not adequate to fully suppress robust overfitting. Besides, comparing the situation of $\lambda > 0$ and $\lambda = 0$ for a fixed $d (\geq 1.5)$, we emphasize that the purposed BoAT loss again exhibits its apparent effectiveness in simultaneously boosting the best robustness and mitigating robust overfitting.

Table 6: Changing decay factor $d$ and regularization strength $\lambda$ and evaluating model robustness on CIFAR-10 under the perturbation norm $\varepsilon_\infty = 8/255$ based on the PreActResNet-18 architecture.

| Method | Natural | | | PGD-20 | | | AutoAttack | | |
|---|---|---|---|---|---|---|---|---|---|
| | best | final | diff | best | final | diff | best | final | diff |
| BoAT($d = 1.3, \lambda = 0.0$) | **81.17** | **81.27** | -0.10 | **56.43** | **56.23** | 0.20 | **50.80** | **50.75** | 0.05 |
| BoAT($d = 1.3, \lambda = 1.0$) | 80.67 | 80.63 | 0.04 | 56.27 | 56.15 | 0.12 | 50.74 | 50.65 | 0.09 |
| BoAT($d = 1.3, \lambda = 4.0$) | 78.50 | 78.36 | **0.14** | 55.10 | 55.04 | **0.06** | 50.31 | 50.30 | **0.01** |
| BoAT($d = 1.5, \lambda = 0.0$) | 82.04 | **82.57** | -0.73 | 56.33 | 56.02 | 0.31 | 50.93 | 50.69 | 0.24 |
| BoAT($d = 1.5, \lambda = 1.0$) | **82.20** | 82.06 | **0.14** | **56.59** | **56.55** | **0.04** | **51.39** | **51.36** | 0.03 |
| BoAT($d = 1.5, \lambda = 4.0$) | 79.68 | 79.88 | -0.20 | 55.72 | 55.65 | 0.07 | 50.52 | 50.50 | **0.02** |
| BoAT($d = 4.0, \lambda = 0.0$) | **83.05** | 85.38 | -2.33 | 55.98 | 50.87 | 5.11 | 50.38 | 46.95 | 3.43 |
| BoAT($d = 4.0, \lambda = 1.0$) | 82.46 | 84.84 | -2.38 | 55.86 | 52.02 | 3.84 | 50.87 | 47.88 | 2.99 |
| BoAT($d = 4.0, \lambda = 4.0$) | 80.99 | 84.07 | -3.08 | **56.06** | **53.58** | **2.48** | **51.00** | **49.02** | **1.98** |

## C.2 Improving Natural Accuracy with Knowledge Distillation

Chen et al. (2021) propose to adopt knowledge distillation (KD) (Hinton et al., 2015) to mitigate RO and it is worth mentioning that their method achieves relatively good natural accuracy according to Table 1 and 2. Since our BoAT method is orthogonal to KD, we propose to combine our techniques with KD to further improve natural accuracy. Specifically, we simplify their method by using only a non-robust standard classifier as a teacher (ST teacher) instead of using both a ST teacher and a AT teacher, because i) a large sum of computational cost for training the AT teacher will be saved, ii) our main goal is to improve natural accuracy so the ST teacher matters more and iii) BoAT already use the WA model as a very good teacher. This gives the final loss function as

$$\ell_{\text{BoAT+KD}}(x, y; \theta) = (1 - \lambda_{\text{ST}}) \cdot \ell_{\text{BoAT}}(x, y; \theta) + \lambda_{\text{ST}} \cdot \text{KL}\left(f_\theta(x) \| f_{\text{ST}}(x)\right), \quad (5)$$

where $f_{\text{ST}}$ indicates the ST teacher and $\lambda_{\text{ST}}$ is a trade-off parameter.

Table 7: Combining our methods with knowledge distillation and evaluating model robustness on CIFAR-10 under the perturbation norm $\varepsilon_\infty = 8/255$ based on the PreActResNet-18 architecture.

| Method | Natural | | | PGD-20 | | | AutoAttack | | |
|---|---|---|---|---|---|---|---|---|---|
| | best | final | diff | best | final | diff | best | final | diff |
| BoAT($\lambda_{\text{ST}} = 0.0$) | 82.20 | 82.06 | **0.14** | **56.59** | **56.55** | **0.04** | **51.39** | **51.36** | 0.03 |
| BoAT+KD($\lambda_{\text{ST}} = 0.4$) | 83.88 | 83.74 | **0.14** | 55.69 | 55.47 | 0.22 | 51.13 | 51.11 | 0.02 |
| BoAT+KD($\lambda_{\text{ST}} = 0.6$) | **84.66** | **84.54** | 0.12 | 54.44 | 54.38 | 0.06 | 50.54 | 50.65 | **-0.11** |

Table 7 compares the performance of BoAT+KD when different $\lambda_{\text{ST}}$ is applied, and clearly a large $\lambda_{\text{ST}}$ results in improvement in natural accuracy and decreases robustness (may due to the theoretically principled trade-off between natural accuracy and robustness (Zhang et al., 2019)). However, it is still noteworthy that when $\lambda_{\text{ST}} = 0.4$, a notable increase in natural accuracy ($\sim 1.70\%$) is achieved at the cost of only a small slide of $\sim 0.25\%$ in robustness against AutoAttack. Moreover, we emphasize that RO is almost completely eliminated regardless of the trade-off, which is the main concern of this paper and demonstrates the superiority of our method against previous ones.

## C.3 Changing Learning Rate Decay Schedule

In the previous experiments we only investigate the piecewise LR decay schedule. However, a natural idea would be using mild LR decay schedules, *e.g.,* Cosine and Linear decay schedule, instead of suddenly decaying it by a factor of $d$ at some epoch in the piecewise decay schedule. Therefore, we experiment with Cosine and Linear decay schedule and summarise the results in Table 8. To be specific, the LR still decays to 0.01 at epoch 150 and 0.001 at epoch 200, though the two decay stages (from epoch 100 to 150 and 150 to 200) are designed to be gradual following the Cosine/Linear schedule. We also gradually increase the strength of BoAT regularization from zero as the LR

gradually decreases, following the "larger decay factor goes with stronger BoAT regularization" principle that we introduced in Appendix C.1.

Table 8: Comparing our method with WA on CIFAR-10 under the perturbation norm $\varepsilon_\infty = 8/255$ based on PreActResNet-18 architecture, when Cosine/Linear LR decay schedule are applied.

| Method | Cosine | | | | | | Linear | | | | | |
|---|---|---|---|---|---|---|---|---|---|---|---|---|
| | Natural | | PGD-20 | | AutoAttack | | Natural | | PGD-20 | | AutoAttack | |
| | best | final | best | final | best | final | best | final | best | final | best | final |
| WA(d=10) | **82.22** | **85.17** | 56.22 | 47.96 | 50.79 | 43.87 | **82.28** | **85.27** | 56.37 | 48.17 | 50.83 | 44.37 |
| **BoAT(d=10)** | 82.16 | 82.17 | **56.24** | **54.38** | **51.01** | **49.76** | 82.03 | 82.03 | **56.53** | **54.67** | **51.17** | **49.63** |

It can be concluded from Table 8 that simply changing the LR decay schedule indeed improves the best robust accuracy from 49.92% to 50.79% and 50.83% against AutoAttack respectively, but it provides no help at all in mitigating RO as the last robust accuracy is still around 44% against AutoAttack. We also note that in this situation, the application of BoAT loss not only significantly mitigates RO but also further improves the best model robustness, which also proves its effectiveness.

## C.4 Using Validation Set

Following Wu et al. (2020); Dong et al. (2022) and Yu et al. (2022), we report the test accuracy on the best checkpoint that achieves the highest robust test accuracy. Another way to evaluate model robustness is selecting the model that achieves the highest robustness on validation set as the best checkpoint. Therefore, following Rice et al. (2020), we hold out 1,000 examples from the CIFAR-10 training set for validation purposes and rerun the experiments using PGD-AT and BoAT.

Table 9: Selecting models of best robustness base on their performance on validation set and evaluating robustness on CIFAR-10 under the perturbation norm $\varepsilon_\infty = 8/255$ based on the PreActResNet-18 architecture.

| Best Model Selection | Method | Natural | | | PGD-20 | | | AutoAttack | | |
|---|---|---|---|---|---|---|---|---|---|---|
| | | best | final | diff | best | final | diff | best | final | diff |
| Using Validation Set | PGD-AT | 81.61 | **84.67** | 3.06 | 52.51 | 46.20 | 6.31 | 47.54 | 42.31 | 5.23 |
| | **BoAT** | **81.86** | 81.91 | **-0.05** | **56.36** | **56.12** | **0.24** | **51.13** | **51.22** | **-0.09** |
| Highest on Test Data | PGD-AT | 82.08 | **83.98** | -1.90 | 52.64 | 47.06 | 5.58 | 47.72 | 42.60 | 5.12 |
| | **BoAT** | **82.20** | 82.06 | **0.14** | **56.59** | **56.55** | **0.04** | **51.39** | **51.36** | **0.03** |

From Table 9, we can see that the selection criterion seems to make little difference. In our BoAT, changing test to validation only causes very slight drop in natural (0.1%) and AA (0.2%) accuracy. Therefore, our validation-based best-epoch robustness (51.13% against AutoAttack) is still significantly better than all the other test-based methods in Table 1.

## C.5 Results on More Network Architectures

In previous experiments we compare methods based on PreActResNet-18 and WideResNet-34-10 architecture, and here we also adopt VGG-16 (Simonyan & Zisserman, 2015) and MobileNetV2 (Sandler et al., 2018) architecture. The significant improvement against baseline PGD-AT in both best and final robust accuracy and in mitigating RO reported in Table 10 further demonstrates that our method is applicable to a wide range of network architectures.

## C.6 Training Efficiency

We also test and report the training time (per epoch) of several methods evaluated in this paper. For a fair comparison, all the compared methods are integrated into a universal training framework and each test runs on a single NVIDIA GeForce RTX 3090 GPU.

From Table 11, we can see that BoAT requires nearly no extra computational cost compared with vanilla PGD-AT (136.2s *v.s.* 131.6s per epoch), implying that it is an efficient training method in practical. We also remark that KD+SWA, one of the most competitive methods that aims to address

Table 10: Comparing our method with PGD-AT on CIFAR-10 under the perturbation norm $\varepsilon_\infty = 8/255$ based on VGG-16 and MobileNetV2 architecture.

| Method | VGG-16 | | | | | | MobileNetV2 | | | | | |
| | Natural | | PGD-20 | | AutoAttack | | Natural | | PGD-20 | | AutoAttack | |
| | best | final | best | final | best | final | best | final | best | final | best | final |
| PGD-AT | **78.75** | **81.95** | 50.76 | 44.58 | 44.56 | 39.96 | **80.12** | **81.03** | 51.56 | 50.67 | 46.01 | 45.27 |
| **BoAT** | 78.37 | 78.65 | **53.66** | **53.41** | **47.65** | **47.55** | 78.98 | 80.81 | **53.18** | **52.57** | **47.66** | **47.35** |

Table 11: Combining training time per epoch on CIFAR-10 under the perturbation norm $\varepsilon_\infty = 8/255$ based on the PreActResNet-18 architecture.

| Method | Training Time per Epoch (s) |
| --- | --- |
| PGD-AT | 131.6 |
| WA | 132.1 |
| KD+SWA | 131.6+16.5+141.7 |
| AWP | 142.8 |
| MLCAT$_{\text{wp}}$ | 353.3 |
| **BoAT** | 136.2 |
| WA+CutMix | 168.6 |
| **BoAT+CutMix** | 173.1 |

the RO issue, is not really computationally efficient as it requires to pretrain a robust classifier and a non-robust one as AT teacher and ST teacher respectively.

## C.7 COMPARISON WITH MLCAT

Recently, Yu et al. (2022) propose their understandings in robust overfitting and propose a method MLCAT to alleviate RO. Here we compare our work with theirs in details.

**Methodologies.** Yu et al. (2022) analyze robust overfitting by investigating the roles of easy and hard samples and propose to regularize samples of small loss values (typically using non-robust features). Nevertheless, their discussions cannot fully explain how robust overfitting rises from easy samples. In comparison, we provide a generic explanation for the rise of robust overfitting from a feature dynamics perspective, and we verify this understanding through extensive experiments. Our proposed BoAT is also designed from a bootstrapping perspective, which is different from theirs. At last, our understandings can also help understand how Yu et al. (2022)'s method works: by regularizing small-loss data, they enhance robust features and diminish non-robust features, which helps alleviate robust overfitting.

Table 12: Comparing our method with MLCAT on CIFAR-10/CIFAR-100 under the perturbation norm $\varepsilon_\infty = 8/255$ based on the PreActResNet-18 architecture. The reported robust accuracy is evaluated with AutoAttack.

| Method | CIFAR-10 | | | CIFAR-100 | | |
| | best | final | diff | best | final | diff |
| --- | --- | --- | --- | --- | --- | --- |
| MLCAT$_{\text{LS}}$ | 28.12 | 26.93 | 1.19 | 13.41 | 11.35 | 2.06 |
| MLCAT$_{\text{WP}}$ | 50.70 | 50.32 | 0.38 | 25.66 | 25.28 | 0.38 |
| **BoAT** | **51.39** | **51.36** | **0.03** | **27.59** | **27.52** | **0.02** |

**Performance and Efficiency.** Table 12 shows that 1) our BoAT method is consistently superior (¿1% robust improvements against AutoAttack) to MLCAT across different architectures and different backbone networks, and 2) BoAT has weaker robust overfitting than MLCAT. Table 11 further shows that 3) BoAT is much more training efficient by requiring only 1/3 training time of MLCAT. Therefore, BoAT is both more effective, more efficient, and has less overfitting than MLCAT.

## D    VISUALIZING ADVERSARIAL EXAMPLES

In Figure 14, We visualize some adversarial examples generated in the experiments (following the settings introduced in Appendix B.3) in Section 2.2 and 2.3.

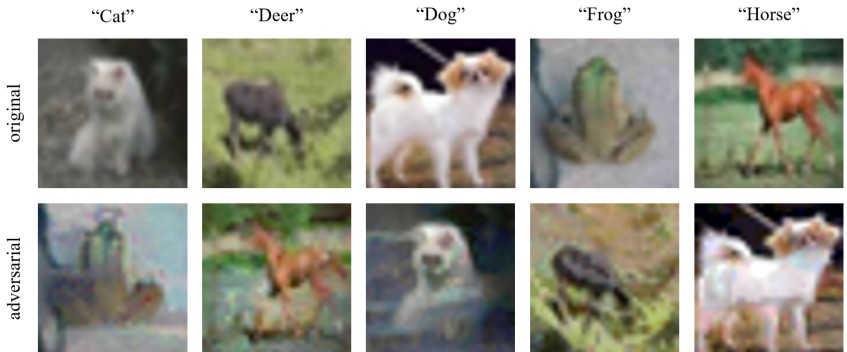

Figure 14: Adversarial examples generated in the experiments in Section 2.2 and 2.3

