# OpenReview forum: "Understanding and Mitigating Robust Overfitting through the Lens of Feature Dynamics"
_ICLR.cc/2023/Conference — Submitted to ICLR 2023_

### Official Review · Reviewer_vLyN · 2022-10-17

**Confidence:** 3
**Correctness:** 3
**Technical Novelty And Significance:** 3
**Empirical Novelty And Significance:** 2
**Recommendation:** 6

**Clarity, Quality, Novelty And Reproducibility:**

## Clarity

The paper is overall clear and well-written. However, the plotting needs to be improved. For example, Figure 2(c) has way too tiny x/y labels. The color in 3(d) is too close for 12/255 and 16/255 to enable one to differentiate one from the other. There are also some typos, for example, in the third to last line of Sec3.3, “under \epsilon = 8/255” seems to be “under \epsilon = 16/255” instead? “purposed BoAT” (end of page 7) should be “proposed BoAT”. Please carefully check the paper for similar errors.


## Quality

The claims regarding the RO after LR decay are mostly supported by the experiments.

## Novelty

The insight of the cause for RO after LR decay is novel. The mitigation strategy leveraging self-supervision is also moderately novel.

## Reproducibility

Although code is not provided, the method seems to be reproducible.


**Strength And Weaknesses:**

## Pros

-neat observations and formulation of the RO problem

-the experiments mostly verify the proposed hypotheses

## Cons

-BoAT has much lower natural accuracy compared with KD+SWA

The results in Table1 seem to suggest BoAT simply trades off natural accuracy and adversarial accuracy compared with existing methods (e.g., KD+SWA) so it might not be very useful in the sense of improving the natural-adversarial accuracy trade-off. I would like to see how the trade-off of natural accuracy and adversarial accuracy look like for different hyper-parameters of BoAT and baselines, and which one lies on the Pareto front.


**Summary Of The Paper:**

This paper studies the robust overfitting (RO) problem for adversarial training (AT) which usually happens after the learning rate (LR) decay. In particular, it hypothesizes that RO is caused by a loss of balance (trainer becomes stronger than attacker) after LR decay. It verifies the hypothesis by looking into target-class information in test robust features, bilateral class correlation, and confusion matrix symmetry for models with and without LR decay. Based on this understanding, the authors propose three techniques (model flatness regularization, smaller LR decay and stronger attacker during training) to rebalance the trainer and the attacker. The proposed method BoAT is evaluated against several baselines on two model architectures and three datasets.

**Summary Of The Review:**

Overall, the paper provides neat observations on the RO problem in AT and verifies the hypotheses. Its proposed three techniques mitigating RO seems to be functional for alleviating RO but might not be useful as general techniques to improve robustness.

---

> ### Author Response · Authors · 2022-11-16
> **Response to Reviewer vLyN**
>
>
> We thank Reviewer vLyN for appreciating the insights, clarity, and effectiveness of our work and for your constructive comments on the writing. We have fixed the plots and typos following your suggestions. Below, we address your main concerns.
>
> ---
>
> **Q1**. BoAT has much lower natural accuracy compared with KD+SWA.
>
> > The results in Table1 seem to suggest BoAT simply trades off natural accuracy and adversarial accuracy compared with existing methods (e.g., KD+SWA) so it might not be very useful in the sense of improving the natural-adversarial accuracy trade-off. I would like to see how the trade-off of natural accuracy and adversarial accuracy look like for different hyper-parameters of BoAT and baselines, and which one lies on the Pareto front.
> >
>
> **A1**. In adversarial training, there is known to be a trade-off in accuracy and robustness, where better robustness often means lower accuracy. The following results (quoted from Table 1) show that our method can **improve both natural accuracy (+1%) and AA robustness (+1%) significantly over previous SOTA (WA+CutMix, Rebuffi et al., 2021)** on a large WRN model. Therefore, our method benefits both natural and robust accuracy, because it suffers much less from overfitting.
>
> *Table 1. Best-epoch Results  on CIFAR-10 with WideResNet-34-10*
>
> | Method | Natural | AA |
>
> |— | — | —|
>
> | PGD-AT (Madry et al., 2018) | 86.09 | 52.11|
>
> | AWP (Wu et al., 2020) | 86.28 | 53.26|
>
> | WA+CutMix (Rebuffi et al., 2021) | 85.41 |55.16|
>
> | **BoAT+CutMix (ours)** | **86.49 (+1.08)** |**56.13  (+0.97)**|
>
> **Further comparison to KD+SWA:**
>
> - Additionally, in Table 1, we also notice that KD+SWA has better natural accuracy than other methods, however, **it still has much lower robustness than our BoAT (e.g. -3% AA on WRN)**. As for why KD+SWA can attain higher natural accuracy than others, we remark that KD+SWA is very different from all the other baselines since it incorporates **an additional static pretrained model for knowledge distillation (KD)**. In other words, ***it is a two-stage method, while the others are end-to-end***. Compared to end-to-end adversarial training, it is relatively easy for KD+SWA to prevent overfitting by simply mimicking a static model. The distillation from an extra model will also help improve its natural accuracy. As quoted above, our method (BoAT+CutMix) can still **achieve both the best natural accuracy and AA robustness among end-to-end methods**.
> - For a further fair comparison with KD+SWA, we also add a KD loss to BoAT using the same pretrained model as KD+SWA, making it also a two-stage method. Results are shown below (details in Appendix C.2). We can see that when further combined with KD, the two-stage BoAT attains both better natural accuracy (+0.06%) and AA robustness (+1.26%) than KD+SWA. It shows that our BoAT is indeed more effective for adversarial training, and it can be further combined with other techniques for attaining better performance.
>
> *Best-epoch Results  on CIFAR-10 with PreActResNet-18*
>
> |  | Natural | AA |
> | --- | --- | --- |
> | KD+SWA (two-stage) | 83.82 | 49.87 |
> | BoAT (end-to-end) | 82.20 | 51.39 |
> | BoAT+KD (two-stage) | 83.88 | 51.13 |
>
> We have added this discussion in **Appendix C.2**.
>
> ---
>
> **Q2**. The plotting needs to be improved. Some typos.
>
> **A2**. We have revised each plot and fixed the typos following your suggestions and the readability indeed improves a lot. Thanks for your advice!
>
> ---
>
> Thanks again for your careful reading and constructive comments. Hope our elaboration and additional experiments could address your concerns. Please let us know if there is more to clarify. Look forward to your reply.

---

> > ### Comment · Reviewer_vLyN · 2022-12-11
> > **Thank you for your detailed responses!**
> >
> > I want to thank the authors for providing very detailed responses! The additional results addressed my initial concern and I will keep my initial assessment.

---

### Official Review · Reviewer_kLGk · 2022-10-20

**Confidence:** 4
**Correctness:** 3
**Technical Novelty And Significance:** 2
**Empirical Novelty And Significance:** 3
**Recommendation:** 6

**Clarity, Quality, Novelty And Reproducibility:**

Clarity is good.

Quality and Novelty are fine.

Reproducibility is unknown

**Strength And Weaknesses:**


Pros

1. The background and motivation of the paper are well-written, and the validation experiments of the proposed hypothesis are convincing.
2. The paper is well-written, clear, and easy to follow.



Cons.

1. The proposed dynamic feature robustness framework needs to be more formally and clearly defined, especially the difference from the former work. For instance, it is encouraged to provide a formal mathematical definition of dynamic, robust, and non-robust features.
2. The mitigation approaches ”adding regularization loss”, “using stronger attacker” and “decreasing learning rate” are well-established tricks and therefore this paper does not bring technical novelty.
3. The proposed methods are only evaluated on 2 architectures in the same model family, which is highly insufficient. More network backbones like MobileNet and VGG should be included.
4. Although the proposed methods lead to improved robustness, it suffers from natural accuracy degradations. It is hard to say whether the BoAT is better compared to other approaches. Also, this needs to be mentioned in the front of this paper, otherwise only mentioning the robustness benefits is misleading.
5. The paper mentions that non-robust features will be much more robust after the learning rate decay. I am very curious about whether the non-robust features will also approach robust features visually.
6. "RO usually happens after the learning rate decay in AT". Actually, this claim is insufficiently supported. Since most previous papers decay the learning rate at the 100th epoch. In order to well support this claim, the authors need to conduct experiments with LR decay at different epochs and show the consistency of RO observations.

**Summary Of The Paper:**

Summary

Based on former findings, this paper first proposes a new definition of robust features with taking dynamics into consideration. Focused on the phenomenon that robust overfitting usually starts after learning rate decay, the authors provide an explanation of robust overfitting based on several experiments. New approaches to mitigate robust overfitting are proposed and validated. The main difference from previous studies is their emphasis on the necessity of considering the min-max nature of AT in robust overfitting.

**Summary Of The Review:**

Good paper with novelty and experiment limitations.

---

> ### Author Response · Authors · 2022-11-16
> **Response to Reviewer kLGk (3/3)**
>
> **Q4**. Although the proposed methods lead to improved robustness, it suffers from natural accuracy degradations. It is hard to say whether the BoAT is better compared to other approaches.
>
> **A4**. In adversarial training, there is known to be a trade-off in accuracy and robustness, where better robustness often means lower accuracy. The following results (quoted from Table 1) show that our method can **improve both natural accuracy (+1%) and AA robustness (+1%) significantly over previous SOTA (WA+CutMix, Rebuffi et al., 2021)** on a large WRN model. Therefore, our method benefits both natural and robust accuracy, because it suffers much less from overfitting.
>
> *Table 1. Best-epoch Results  on CIFAR-10 with WideResNet-34-10*
>
> | Method | Natural | AA |
> | -- | -- | --|
> | PGD-AT (Madry et al., 2018) | 86.09 | 52.11|
> | AWP (Wu et al., 2020) | 86.28 | 53.26|
> | WA+CutMix (Rebuffi et al., 2021) | 85.41 |55.16|
> | **BoAT+CutMix (ours)** | **86.49 (+1.08)** |**56.13  (+0.97)**|
>
> **Further comparison to KD+SWA:**
>
> - Additionally, in Table 1, we also notice that KD+SWA has better natural accuracy than other methods, however, **it still has much lower robustness than our BoAT (e.g. -3% AA on WRN)**. As for why KD+SWA can attain higher natural accuracy than others, we remark that KD+SWA is very different from all the other baselines since it incorporates **an additional static pretrained model for knowledge distillation (KD)**. In other words, ***it is a two-stage method, while the others are end-to-end***. Compared to end-to-end adversarial training, it is relatively easy for KD+SWA to prevent overfitting by simply mimicking a static model. The distillation from an extra model will also help improve its natural accuracy. As quoted above, our method (BoAT+CutMix) can still **achieve both the best natural accuracy and AA robustness among end-to-end methods**.
> - For a further fair comparison with KD+SWA, we also add a KD loss to BoAT using the same pretrained model as KD+SWA, making it also a two-stage method. Results are shown below (details in Appendix C.2). We can see that when further combined with KD, the two-stage BoAT attains both better natural accuracy (+0.06%) and AA robustness (+1.26%) than KD+SWA. It shows that our BoAT is indeed more effective for adversarial training, and it can be further combined with other techniques for attaining better performance.
>
> *Best-epoch Results  on CIFAR-10 with PreActResNet-18*
>
> |  | Natural | AA |
> | --- | --- | --- |
> | KD+SWA (two-stage) | 83.82 | 49.87 |
> | BoAT (end-to-end) | 82.20 | 51.39 |
> | BoAT+KD (two-stage) | 83.88 | 51.13 |
>
> We have added this discussion in **Appendix C.2**.
>
> ---
>
> **Q5**. The paper mentions that non-robust features will be much more robust after the learning rate decay. I am very curious about whether the non-robust features will also approach robust features visually.
>
> **A5**. We are afraid that you might misunderstand something here. Here, we are referring the robustness change of the **same group of features.** In our definition, **the robustness of a feature is not static, and not solely a data property**, instead, it could be changing along training, **depending on both the trainer and the attacker**. Therefore, we are referring to **the same group of features** when we say they change from non-robust to robust. For example, in the experiment in Fig 1c, we are testing the same group of input images **(therefore, nothing changes visually, see the added Figure 14 in Appendix D)**, while their robustness (dependent on the model) indeed changes during the training process.
>
> ---
>
> **Q6**. The claim "RO usually happens after the learning rate decay in AT" is insufficiently supported. In order to well support this claim, the authors need to conduct experiments with LR decay at different epochs and show the consistency of RO observations.
>
> **A6.** For completeness, we also evaluate the situations where LR decays at different epochs (25, 50, 75, 100, 125, 150, 175). Results are shown in **Figure 7 (Appendix B.2)**. We can see that wherever the decay happens, test robustness will **always** **start to decrease (i.e., overfit) shortly after the decay**, implying that our claim "RO happens after the learning rate decay in AT" indeed holds in general cases. We added these results in **Appendix B.2** to support our claims following your suggestion.
>
> ---
>
> Thanks for your insightful and constructive comments. Hope that our additional experiments and explanations above could address your concerns. Please let us know if you have additional questions. We are willing to take them during the rebuttal stage. Look forward to your reply.

---

> ### Author Response · Authors · 2022-11-16
> **Response to Reviewer kLGk (2/3)**
>
>
> **Q3**. More network backbones like MobileNet and VGG should be included.
>
> **A3**. Though previous related works rarely conduct experiments on network architectures beyond (PreAct)ResNet and WideResNet, we would like to follow your suggestion and evaluate BoAT on CIFAR-10 based on VGG-16 & MobileNetV2 architecture when the perturbation norm is 8/255.
>
> *Results of VGG-16  on CIFAR-10*
>
> | PGD-AT | natural | PGD-20 | AA |
> | --- | --- | --- | --- |
> | best | 78.75 | 50.76 | 44.56 |
> | last | 81.95 | 44.58 | 39.96 |
> | BoAT (ours) | natural | PGD-20 | AA |
> | best | 78.37 | 53.66 | 47.65 |
> | last | 78.65 | 53.41 | 47.55 |
>
> *Results of MobileNetV2 on CIFAR-10*
>
> | PGD-AT | natural | PGD-20 | AA |
> | --- | --- | --- | --- |
> | best | 80.12 | 51.56 | 46.01 |
> | last | 81.03 | 50.67 | 45.27 |
> | BoAT (ours) | natural | PGD-20 | AA |
> | best | 78.98 | 53.18 | 47.66 |
> | last | 80.81 | 52.57 | 47.35 |
>
> We can see that our BoAT can obtain significant improvement against PGD-AT in both best and final robust accuracy and mitigate robust overfitting. further demonstrates that our method is applicable to a wide range of network architectures. We added this experiment to Appendix C.5.

---

> ### Author Response · Authors · 2022-11-16
> **Response to Reviewer kLGk (1/3)**
>
>
> We thank Reviewer kLGk for appreciating the clarity and effectiveness of our work. Below, we address your main concerns.
>
> ---
>
> **Q1**. The proposed dynamic feature robustness framework needs to be more formally and clearly defined, especially the difference from the former work. For instance, it is encouraged to provide a formal mathematical definition of dynamic, robust, and non-robust features.
>
> **A1**.  **Formal difference to Ilyas et al.** The **formal differences** between our framework and Ilyas et al. lie in **the formal definition of robustness $R(f)$** in eq. 2 & 3, respectively. Specifically, in Ilyas et al., their robustness $R(f)$ is static (only dependent on attacker $\Delta$ and distribution $\mathcal{P}$), while in our definition, the robustness $R(f)$ is dynamic as it depends on three factors: attacker $\Delta$, trainer $\mathcal{T}$, and dataset $\mathcal{D}$. Specifically, when the trainer $\mathcal{T}$ changes (e.g. LR decays), the robustness may also vary and become dynamic. This definition gives a formal and general characterization of our feature robustness.
>
> **Formal definition of robust and non-robust features.** As mentioned in the **footnote at page 3**, we formally decide whether a feature $f$ is robust or non-robust feature based on a given threshold $\gamma$. Specifically, we call a feature as robust feature if $R(f)\geq\gamma$ and non-robust otherwise. It is a clear and formal definition, and we have revised the statement to be more clear on this point.
>
> **Dynamic features? Dynamic robustness!** Besides, we note that our framework does **not** regard features as dynamic (features are independent of models in our definition in eq. 3). Instead, it is the **robustness of a feature that** **dynamically changes w.r.t. model training**. This is formally defined through the dependence of feature robustness $R_{\Delta,\mathcal{T},\mathcal{D}}$ (eq. 3) on the attacker $\Delta$, the trainer $\mathcal{T}$, and the dataset $\mathcal{D}$. Therefore, as long as any of the three factors change along training, the feature robustness will vary.
>
> To see how the robustness of a feature may vary, we also give a simple example here. Consider a two-dimensional mixture of Gaussian data of two classes, where data from class 1 follow $N((1,0), I)$, and data from class -1  follow $N((-1,0),I)$.  We can else adopt a linear classifier and an MLP classifier to fit this data, and they will learn different decision boundaries. Then, when utilizing the x-axis feature of the data for attack (perturbing along the direction (1,0)), the attacker will achieve different success rates under two models. Therefore, the same x-axis data feature could have very different robustness under two different models. Similarly, a feature can be robust under a weak attacker (smaller epsilon, less steps), and non-robust against a strong attacker. These evidences show that changing trainer or attacker will result in a dynamic change of feature robustness.
>
>
> ---
>
> **Q2**. The mitigation approaches ”adding regularization loss”, “using stronger attacker” and “decreasing learning rate” are well-established tricks and therefore this paper does not bring technical novelty.
>
> **A2**. We respectfully disagree, and we have summarized the main differences to previous methods in **Section 5.** Below,  we highlight some key points in each aspect:
>
> - **On Stronger Attacker**. Remarkably, among the three tricks, **we are the first to explore “using stronger attacker”**. As mentioned in Sec 5 (3rd paragraph), all previous methods only explored **adopting a weaker attacker** at the beginning stage, but ***none of them adopted a stronger attacker for countering robust overfitting***.
> - **On Regularization**. Regularization loss is a common technique utilized in many well-known AT methods. Therefore, adopting the regularization approach does not make our method lack of novelty, and the key is to design a proper regularization from new perspectives. In this work, we are **the first to adopt a bootstrapping** method for regularizing the flatness of online models, and it is **the first method** **to solve the robust overfitting of Weight Average (WA) methods**. Since WA is now the SOTA method for AT (see RobustBench), we believe that our proposed method does push the boundary of AT.
> - **On LR decay**. Although there are extensive studies on LR schedules, **how it affects adversarial robustness and in particular, robust overfitting, is yet unclear to the community**. In this paper, we are the first attempt to understand the inherent relationship between LR decay and robust overfitting. Based on our understanding, we show that we can largely mitigate RO by simply adjusting the decay factor, which is also a new message to the community.
>
> The discussions above show both our three explorations bring new perspectives, new techniques, and real benefits to the community.  Therefore, we believe that our proposed method does bring technical contributions.

---

> ### Comment · Reviewer_kLGk · 2022-12-10
> **Response to the rebuttal**
>
> Many thanks for all the great efforts during the rebuttal, especially for the newly added experiments. Most of my concerns are addressed.
>
> I remain slightly concerned about the technical novelty. "The discussions above show both our three explorations bring new perspectives, new techniques, and real benefits to the community." Hope the authors can include some discussion in the conclusion section of the final version, and point out how the proposed techniques can inspire new research in this community. Thanks!
>
> I will keep my score as weakly accept: 6.

---

> > ### Author Response · Authors · 2022-12-10
> > **Thanks and we will add discussions to the conclusion section**
> >
> > Thanks for your reply and for appreciating our response. We are glad to hear that our newly added experiments address most of your concerns. Your constructive comments indeed make our results more solid. Following your suggestions, we will add the discussions on the technical novelty (elaborated in A2 in the previous reply) in the revision.
> >
> > As for how the proposed techniques will inspire new research, our proposed new understandings of robust overfitting (RO) through the dynamic balance between attacker and model trainer could serve as a guideline for future research. Generally, to avoid possible RO, they should pay careful attention to striking a balance between the two players. Technically, our perspective provides a unified understanding of existing approaches for avoiding RO (Section 5), which could further inspire the design of more goal-oriented  methods.
> >
> > We give two possible ideas for future works for illustration. First, previously, we only know that strong data augmentations can help prevent RO, but do not know why. Now, as our theory suggests that the role of data augmentations here is mainly to corrupt non-robust features (Section 5), we could design more specific / adaptive data preprocessing / augmentation scheme focusing directly on corrupting non-robust features, instead of relying on hand-designed augmentations. Second, we now still rely on a manually tuned schedule to balance the two players, which is a hard task. In the future, we could design an automatic / learnable schedule for dynamically adjusting the two players according to the training process.
> >
> > Therefore, we believe that our new perspective of AT could be a fruitful source of inspirations for designing better AT algorithms. Hope this explanation could address your concerns, and thanks again for your reply.

---

### Official Review · Reviewer_p7yt · 2022-10-20

**Confidence:** 2
**Correctness:** 1
**Technical Novelty And Significance:** 2
**Empirical Novelty And Significance:** 2
**Recommendation:** 3

**Clarity, Quality, Novelty And Reproducibility:**

No code is provided and the paper is not really clear to me. I don't think I will be able reproduce this work.

**Strength And Weaknesses:**

Weakness:
- The first paragraph in page 5 is confusing.
  - In Figure 2, there are no measurement on feature, only on classes. But the conclusion is on target-class features. What is the definition of non-robust feature?
  - The relationship between the confusion matrix being symmetric and robust overfitting is unclear.
  - What does A->B and B->A mean in Figure 2?
  - What does accuracy with misclassified labels in Figure 2a mean?
  - Is the confusion matrix computed with adversarial attack?
- Some other parts of the paper that are confusing:
  - Should the output of f be a feature or a label? In the text, it's called feature while in equation 2 and 3, they are compared with label y.
  - The authors claimed "False Non-robust Mapping Opens Shortcuts for Test-time Adversarial Attack". What is a shortcut? Could you measure this "shortcut"? Are there any hypothesis on why this shortcut is related to LR decay?
- What is the learning rate schedule used in Figure 2? Does different scheduling or decay method matters?
- The method used in [1] should also be compared. Their method is effectively make the model locally smooth, which in some way also improves landscape flatness.
- For Equation (4), is the adversarial example computed on the cross-entropy loss or on the BoAT loss? The inner minimization should be included in the equation to make things more clear.
- How is the lambda in Equation (4) chosen?
- This paper discussed a lot on robust overfitting, while in section 3 and 4, there is not discussion on the generalization gap comparing with other methods.
- The hyper parameter choices in Table 4 seems arbitrary.
- I think it is arguable whether adversarial training is SOTA algorithm for adversarial robust training. I would suggest checking out some of the methods in https://robustbench.github.io/
- In the paragraph before Section 4.1: "We train each model for 200 epochs and record the best-epoch (selected based on PGD attack)and final-epoch robust test accuracy.". One should not just select the best epoch on test data. This should be done on a validation set.

[1] Zhang, Hongyang, et al. "Theoretically principled trade-off between robustness and accuracy." International conference on machine learning. PMLR, 2019.


**Summary Of The Paper:**

This paper begins with an argument that the robustness of a model should also be conditioned on the training process. They then focus on  the one specific aspect of the training process -- learning rate (LR) decays. They found that after LR decays, the confusion matrix becomes more symmetric. The author claimed that this observation helps to understand robust overfitting. Then, they proposed the BoAT loss, which seems to perform better than some prior work in their experiment.

**Summary Of The Review:**

This paper is not well written. The flow of the paper is also not logical (e.g. the relationship between the Symmetrized Confusion Matrix and robust overfitting is not explained). There are major flaws in the experiment setup (e.g. selecting the best epoch based on the performance on test data), and the experimental result cannot support all their claims.

---

> ### Author Response · Authors · 2022-11-16
> **Response to Reviewer p7yt (4/4)**
>
> **Q8**. The hyper parameter choices in Table 4 seems arbitrary.
>
> **A8**. As revealed in our theory, a non-overfitting adversarial training process requires **a careful** **balance between three things at the same time**: 1) data and augmentation property (decides the amount of non-robust features); 2) the model capacity (decides its initial fitting ability), 2) the LR schedule (decides the fitting ability after the decay).
>
> Therefore, **our choice of the two hyperparameters** (decay factor $\lambda$ and regularization coefficient $\lambda$) **is designed to balance the three elements above** with the following three guiding principles:
>
> - 1) complex data (like Tiny ImagNet) and strong data augmentations (like CutMix) allow a larger $d$;
> - 2) larger models (like WideResNet) require a smaller $d$ to avoid overfitting;
> - 3) stronger model regularizations (larger $\lambda$) are needed to counter stronger fitting ability brought by larger LR decay factor, and vice versa.
>
> You can find that **the hyperparameters in Table 4 all follow our principles above.** Therefore, it is ***not* an arbitrary choice**.
>
> ---
>
> **Q9**. I think it is arguable whether adversarial training is SOTA algorithm for adversarial robust training. I would suggest checking out some of the methods in [https://robustbench.github.io/](https://robustbench.github.io/)
>
> **A9**. The word “adversarial training” is often utilized to refer to the general strategy that improves robustness using in a minimax scheme (inner loop for generating adversarial examples, and outer loop for model training), and not specifically Madry’s method (which we denoted as PGD-AT). In this sense, all SOTA methods on RobustBench are adversarial training methods.
>
> ---
>
> **Q10**. There are major flaws in the experiment setup (e.g. selecting the best epoch based on the performance on test data).
>
> **A10**. We note that it is **a common practice among adversarial training methods to use test datasets for selecting best checkpoints, as adopted in TRADES, PGD-AT+TE, and AWP**. For a convenient comparison with these prior works, we also adopt the same practice as theirs.
>
> Meanwhile, to further address your concerns, we also conduct experiments using validation-based early stopping, where we create a held-out validation set with 1,000 samples from the training data. A comparison between PGD-AT and BoAT with test and validation early stopping is listed below.
>
> | Early-stop criterion | Method | Natural (best) | Natural (last) | AA (best) | AA (last) |
> | --- | --- | --- | --- | --- | --- |
> | test | PGD-AT | 82.08 | 83.98 | 47.72 | 42.60 |
> |  | BoAT (ours) | 82.20 | 82.06 | 51.39 | 51.36 |
> | validation | PGD-AT | 81.61 | 84.67 | 47.54 | 42.31 |
> |  | BoAT (ours) | 81.86 | 81.91 | 51.13 | 51.22 |
>
> We can see that the **selection criterion seems to make little difference**. In our BoAT, changing test to validation **only causes a very slight drop in natural (0.1) and AA (0.2) accuracy.** Therefore, our validation-based best-epoch robustness (51.13 AA) is **still significantly better than all the other test-based methods** in Table 1 (the 2nd is AWP with 50.34 AA).
>
> Besides, we can see that when using validation-based early stopping, the last-epoch AA is even better than best-epoch AA in our BoAT, showing that ***early stopping is indeed not necessary for our method***, and our method indeed successfully mitigates robust overfitting.
>
> ---
>
> Thanks again for your constructive comments. Hope our additional explanations and results above could address your concerns. Please let us know if there is more to clarify. We are looking forward to your reply and willing to take your further questions during the rebuttal stage.

---

> ### Author Response · Authors · 2022-11-16
> **Response to Reviewer p7yt (3/4)**
>
>
> **Q3**. What is the learning rate schedule used in Figure 2? Does different scheduling or decay method matters?
>
> **A3**. As introduced in **Appendix B.1**, all the experiments in Section 2.2 and 2.3 adopt **the canonical piecewise LR decay schedule**, with a decay factor of 10 at the 100-th and the 150-pth epoch.
>
> Following your suggestion, we also experiment with different LR schedules. The results are included in **Figures 10 and 11 (Appendix B.3)**. We can see that i) simply changing LR schedules cannot prevent RO from happening (same results as Rice et al [1]) and **ii) our three empirical verifications on our understandings in the cause of RO still holds** (the test adversarial examples indeed contain less and less target-class information as the training goes; the bilateral class correlation becomes increasingly strong; the confusion matrix indeed becomes symmetric).
>
> The results are almost the same as the results achieved when we adopt piecewise LR decay schedule, indicating that our understandings of the cause of RO is fundamental **regardless of whatever LR decay schedule is used**.
>
> Reference:
>
> [1] Rice, Leslie, Eric Wong, and Zico Kolter. Overfitting in adversarially robust deep learning. In ICML 2020.
>
> ---
>
> **Q4**. TRADES should also be compared.
>
> **A4**. Following your suggestion, we have run the experiments using TRADES and added the results into Table 1 and Table 2 (quoted below).
> #### *Comparison with TRADES on CIFAR-10 (Table 1)*
>
> | Model | Method | best natural | last natural | best AA | last AA |
> | --- | --- | --- | --- | --- | --- |
> | ResNet-18 | TRADES | 80.72 | 82.61 | 48.37 | 46.94 |
> |  | BoAT (ours) | 82.20 | 82.06 | 51.39 | 51.36 |
> | WideResNet-34-10 | TRADES | 84.73 | 84.62 | 53.25 | 45.29 |
> |  | BoAT (ours) | 85.21 | 85.41 | 54.94 | 54.69 |
> |  | BoAT+CutMix(ours) | 86.49 | 86.86 | 56.13 | 56.23 |
>
> #### *Comparison with TRADES on CIFAR-100 and Tiny-ImageNet (Table 2)*
>
> | Dataset | Method | best natural | last natural | best AA | last AA |
> | --- | --- | --- | --- | --- | --- |
> | CIFAR-100 | TRADES | 57.09 | 55.53 | 24.51 | 22.86 |
> |  | BoAT (ours) | 56.44 | 56.51 | 27.59 | 27.52 |
> | Tiny-ImageNet | TRADES | 47.99 | 47.79 | 16.66 | 16.29 |
> |  | BoAT (ours) | 47.79 | 48.20 | 20.41 | 19.88 |
>
> We can see that BoAT has s**ignificant robustness than TRADES (e.g., 48.37 → 51.39) and much less overfitting (e.g., 45.29→56.23)** across different backbone networks, and different datasets. On CIFAR-10, BoAT even has better natural accuracy (~1.5%↑) than TRADES. These results show that BoAT indeed has much better robustness than TRADES while suffering much less from robust overfitting.
>
> ---
>
> **Q5**. For Equation (4), is the adversarial example computed on the cross-entropy loss or on the BoAT loss? The inner minimization should be included in the equation to make things more clear.
>
> **A5**. It is computed on the cross-entropy loss as in PGD-AT. We have added this explanation in the revision following your suggestion.
>
> ---
>
> **Q6**. How is the lambda in Equation (4) chosen?
>
> **A6**. **Following TRADES** (which also has a lambda in their objective), we also select the best lambda according to the performance on test data. We note that this is a common practice among existing AT methods, including the baselines listed in Table 1. We follow this common protocol for a fair comparison.
>
> ---
>
> Q7. This paper discussed a lot on robust overfitting, while in section 3 and 4, there is no discussion on the generalization gap comparing with other methods.
>
> A7. As the title suggests, the main focus of this work is to understand and mitigate the robust overfitting (RO) phenomenon. Meanwhile, we also find that mitigating RO with our BoAT method also helps a lot in closing the generalization gap.
>
> Specifically,  we add a new plot comparing the generalization gap of PGD-AT and BoAT in **Figure 13 (Appendix B.6)**. As quoted below, our method not only mitigates RO at the 200-th epoch by **improving 9% test robustness**, but also significantly **closes the robust generalization gap from 35% to 9%**. Specifically, in BoAT, the train robustness is successfully suppressed (82%→65%) with strong model regularization, and as analyzed in our theory, this also helps improve the test robustness by preventing RO.
>
> #### *Robust generalization gap at the 200-th epoch*
>
> | Method | Train (PGD-10) | Test (PGD-10) | Train-test Gap |
> | --- | --- | --- | --- |
> | PGD-AT | 82% | 47% | 35% |
> | BoAT (ours) | 65% | 56% | 9% |

---

> ### Author Response · Authors · 2022-11-16
> **Response to Reviewer p7yt (2/4)**
>
> **Q2**. Some other parts of the paper that are confusing:
>
> **A2**. We address each of your concerns point by point below.
>
> > Should the output of f be a feature or a label? In the text, it's called feature while in equation 2 and 3, they are compared with label y.
> >
>
> In our definition (Eq. 3), **the feature g** is a mapping from the input space X to the input space X, while its robustness depends on **all classifiers $f\in\mathcal{F}^\mathcal{T}_g$ that** map from the input space X to the label space Y. Ilyas et al (Eq. 2) indeed regard such a classifier f as a feature (see their Section 2 for an explanation). Different from them, we do not regard f as a feature, ****but use only its prediction as an indicator for the belonging class of the feature g.
>
> > The authors claimed "False Non-robust Mapping Opens Shortcuts for Test-time Adversarial Attack". What is a shortcut? Could you measure this "shortcut"? Are there any hypothesis on why this shortcut is related to LR decay?
> >
> - **1) What is the shortcut?** As **explained in the paragraph following this claim** (Sec 2.3 2nd paragraph), after LR decay, the false non-robust mapping will become a shortcut for the attacker, in the sense that it makes it easier for the attacker to succeed. Specifically, the attacker could now cause misclassification by **adding non-robust features from its original class**, which is easier than adding features from a different target class (farther in feature space).
> - **2) How to measure the shortcut?** We measure this shortcut in Experiment 1 (Fig 2a) by showing that **there is indeed less target-class information in the adversarial examples** generated after LR decay. Experiments 2 and 3 also help verify this process, as we explained in the paper.
> - **3) Why LR decay leads to this shortcut?** The reasoning shows that the existence of this shortcut is due to **the false non-robust mappings induced by LR decay.** As we have analyzed in **Sec 2.2 (2nd paragraph),** Li et al [1] pointed out that *“small LR often favors easy-to-generalize and hard-to-fit patterns”*, which correspond to non-robust features in adversarial training. Therefore, as we further analyzed in **Sec 2.3 (2nd paragraph)**, “after LR decay, the trainer T becomes more capable of memorization with a smaller LR”, and these memorized non-robust features act as false non-robust mappings. Overall, we have explained how LR leads to the shortcut through the reasoning `LR decay -> false non-robust mapping -> shortcut` . Please refer to the paper for more detailed elaborations.
>
> We have revised the discussion here to better explain this relationship. Hope you find the explanation above satisfactory.
>
> Reference:
>
> [1] Yuanzhi Li, Colin Wei, and Tengyu Ma. Towards explaining the regularization effect of initial large learning rate in training neural networks. In NeurIPS, 2019.

---

> ### Author Response · Authors · 2022-11-16
> **Response to Reviewer p7yt (1/4)**
>
> We thank Reviewer p7yt for your time and efforts on reviewing our work. Below, we provide a detailed response to each of your concerns.
>
> ---
>
> Q1. The first paragraph in page 5 is confusing.
>
> **A1**. We address each of your concerns point by point below.
>
> > In Figure 2, there are no measurement on feature, only on classes. But the conclusion is on target-class features. What is the definition of non-robust feature?
> >
>
> As shown in our definition (eq. 3), features should be useful for classification, and their robustness depends on the sensitivity to perturbations. In the classification task (considered in AT), only class-related features (e.g., dogs’ ears and car’s tiers) will be utilized in the classifier to tell different classes apart. Therefore, **the belonging classes largely indicate the property of features**. Therefore, in Figure 2, the change of accuracy indicates the shift of features utilized in the classifier. Since it is hard to tell what features are exactly used in the classifier, both Ilyas et al and ours the classification accuracy for a quantitative measure of the change of features.、
>
> > What does A->B and B->A mean in Figure 2?
> >
>
> This stems from our illustrative discussions in the third paragraph of Sec 2.3, where we take an
> adversarial example x from class A that is misclassified to class B as an example, and analyze how it overfits during the LR decay. This example is used throughout the entire analysis in Section 2.3, from the explanation part to the experiment part.
>
> > The relationship between the confusion matrix being symmetric and robust overfitting is unclear.
> >
>
> In the third paragraph in Sec 2.3, we have explained that the robust overfitting phenomenon arises when a test sample from class B is attacked to class A with a shortcut g, a non-robust feature from the same class (B). Because of this shortcut, we will have a deduction that class A→B misclassification will induce a B→A misclassification (explained in Experiment 2), which will naturally lead to the symmetry of the confusion matrix (Experiment 3). Therefore, the verification of this logical deduction (symmetry of confusion matrix) will help verify our explanation of robust overfitting. Reviewer yb7Z mentioned that the symmetrized confusion matrix is novel, interesting, and convincing, and regard it as a major strength of this work.
>
> Therefore, we are afraid that we do not fully get your questions here. Please let us know **which part of the deduction above is unclear to you, or what specific logical or factual errors it has**. We would be very willing to further explain it to you.
>
> > What does accuracy with misclassified labels in Figure 2a mean?
> >
>
> It means the accuracy computed w.r.t. **the class labels that these adversarial examples are misclassified to**. For example, generally speaking, if a sample x from class A is misclassified to class B (the target class), then its non-robust features are now supposed to be **class B (i.e., the misclassified label)**. But in our analysis, if the shortcut effect exists (explained in Sec 2.3), many should be still about its original class A. To verify this, we compute the accuracy between predicted class of this non-robust feature and class B. Figure 2a shows that the accuracy keeps going down after LR decay, showing that more and more non-robust features injected during overfitting indeed do not come from the target class B. This thus help verify our analysis of shortcut effects. Experiment details are included in Appendix B.2, and we have added references to it.
>
> > Is the confusion matrix computed with adversarial attack?
>
> Yes, we compute confusion matrices over adversarial examples generated under PGD-20 attack.
>
> Overall, we hope our explanations above could ease your doubts. For better clarity, we have revised the caption of Figure 2 and add references to the discussion in Sec 2.3 following your suggestions.

---

> ### Comment · Reviewer_p7yt · 2022-12-12
> **Response**
>
> Sorry for leaving my previous response in the wrong section, you can find it [here](https://openreview.net/forum?id=0JD3EN75NJE&noteId=NfUzmkufjTS).
>
> Thank you for responding to my response. After reading the response, I think that the relationship between LR decay, dynamic feature, and the proposed algorithm is still unclear. I would suggest presenting a concrete example (even on a toy dataset) of what kinds of new robust features are learned and how these features appear only with LR decay but not other regularization methods. Strengthening the connection between BoAT and dynamic features is also a good idea. For the SOTA claim part, I agree with another comment that toning it down would be better. I would also suggest using the result under validation-based early stopping in the main paper.
>
> Based on these, I decide to keep my original decision.

---

> > ### Author Response · Authors · 2022-12-13
> > **Thanks and Further Response**
> >
> > Thanks for your response. We address your further concerns below.
> >
> > ---
> > **Q1.** The relationship between LR decay and dynamic feature.
> >
> > **A1.** In the [A1 of the last reply](https://openreview.net/forum?id=0JD3EN75NJE&noteId=u2gClp5Swei), we have explained how LR decays have a direct influence on overfitting (Figure 7, Appendix B.2). Our dynamic feature robustness framework is proposed to explain this phenomenon. We note that it is generally hard to quantitatively measure the regularization effect of LR decay. This problem is a hard problem even for the deep learning theory community, since we all know that different LR has a large effect on NN performance, but there is hardly concrete theoretical justification for this phenomenon. Therefore, we mainly resort to empirical justifications in this paper, and the measured change of feature robustness does support our theory.
> >
> > ---
> > **Q2.** The connection between dynamic feature and the proposed method.
> >
> > **A2.** In fact, we have elaborated in detail (Section 3) that how we motivate our method from the perspective of dynamic feature robustness. Specifically, the BoAT objective is designed to enforce a flatter landscape to avoid overfitting to non-robust features after LR decay, and the smaller LR decay rate is designed to limit the models' local fitting ability to non-robust features. In Section 5, we also draw a connection between existing approaches to prevent RO and our dynamic perspective.  Therefore, our perspective can provide guidance for practical designs and our method is a direct solution to alleviating non-robust features following our theory.
> >
> > ---
> > Hope the explanation above could address your concerns. Please let us know if there is more to clarify.

---

### Official Review · Reviewer_yb7Z · 2022-10-25

**Confidence:** 4
**Clarity, Quality, Novelty And Reproducibility:** Good clarity, quality, novelty, and r…
**Correctness:** 3
**Technical Novelty And Significance:** 3
**Empirical Novelty And Significance:** 3
**Recommendation:** 8

**Details Of Ethics Concerns:**

None.

**Strength And Weaknesses:**

Strength:
- The considered symmetrized confusion matrix is interesting and convincing, which is novel to me.
- The designed objective function is inspiring. Specifically, the authors introduce a KL term, minimizing the KL divergence between the learnable model f_{\theta} and the detached model f_{\phi}, which is actually equal to assigning a soft label to each adversarial example. Thus, it is promising and exciting to explore a novel approach to designing different assignments of soft labels. And, maybe, it has potential benefit to the research of certifiable defense.
- The paper is well-written, easy to follow, and has solid experimental analysis.
- The proposed method is simple yet effective, verified by the experiments.

Weakness:
- A stronger attack for stronger defense is not employed due to its harmful impacts on natural accuracy, but I cannot find any explanations about the observation from the proposed perspective. It is necessary to add the corresponding explanation using the proposed view because methods inspired by the proposed framework are dropped from the main page, making the framework less convincing.
- The definition of Eq. (3) seems confusing. The trainer T is directly applied to update the model, i.e., changing the learned representations, thus, introducing the trainer T, given the representation, seems meaningless, but the trainer T is important for the rest of the work. Hence, the authors should clarify the point. Similarly, the utilized F_f^T is also confusing.

Suggestions:
1 The claimed theory seems to be a hypothesis, as merely experimental analysis is given in the current version.

2 According to the claimed perspective -- the main goal is to prevent the model trainer T from fitting non-robust features too quickly and too adequately --, a straightforward approach can be that the trainer decreases the learning rate progressively from the previous epoch decaying the learning rate by 0.1. Thus, the corresponding experiments are required. However, it is lacking in the current version.

3 Different from the section on the method, the section on explanation seems weak, I suggest employing more references to convince the claim, the related perspective is the spurious feature perspective, such as CausalADV and group DRO.

4 The ‘feature’ considered in Eq. (2) corresponds to the data space, while the ‘feature’ considered in this work Eq. (3) is more about the representation space, thus, I suggest the authors make these two features distinguishable as they are conceptually different.

[CausalADV] Adversarial robustness through the lens of causality. Zhang et al. 2021

[group DRO] Distributionally robust neural networks for group shifts: On the importance of regularization for worst-case generalization. Sagawa et al., 2020


**Summary Of The Paper:**

This work aims to provide a new perspective to explain the robust overfitting and improve the adversarial robustness built upon the proposed viewpoint. To do so, the authors revisit and generalize the static framework of feature robustness. Built upon the proposed framework, the authors claim that the balance between the attacker and the trainer is the key to understanding robust overfitting. Consequently, the authors propose three methods inspired by the framework and demonstrate the effectiveness of these (two, as one is dropped in the main page) methods.

**Summary Of The Review:**

The proposed framework is novel to me, the proposed methods are simple yet effective, and the conducted experiments are convincing.

---

> ### Author Response · Authors · 2022-11-16
> **Response to Reviewer yb7Z (2/2)**
>
> **Q6.** The ‘feature’ considered in Eq. (2) corresponds to the data space, while the ‘feature’ considered in this work Eq. (3) is more about the representation space, thus, I suggest the authors make these two features distinguishable as they are conceptually different.
>
> **A6.** We are afraid there are some misunderstandings here. In Eq. (3), we defined a feature g as a filter $g:\mathbb{X}\to\mathbb{S}$, where $\mathbb{S}$ denotes the feature space. Features here still refer to the canonical concepts of features (such as shape, texture) in the data space, not in the representation space of neural networks.  For clarity, we now define $g:\mathbb{X}\to\mathbb{X}$ as an input space transformation in the revision (**Section 2.2**).
>
> ---
>
> We thank Reviewer yb7Z for your careful reading and constructive comments, which helps us improve the clarity and rigorism of this work. Hope you find our explanations and revisions above satisfactory. We would be happy to your additional questions during the rebuttal stage.

---

> > ### Comment · Reviewer_yb7Z · 2022-11-21
> > **Thank you for the response and the corresponding extensive revision**
> >
> > I truly appreciate the efforts the authors have made. The authors' excellent rebuttal addresses all my concerns, especially, the experiments mentioned in A4 make the results more convincing.  Accordingly, I have improved my score to 8.

---

> > > ### Author Response · Authors · 2022-12-10
> > > **Thanks for appreciating our response**
> > >
> > > Thanks very much for your encouraging reply and for appreciating our response. We are very glad to hear that we have addressed all your concerns, and your constructive comments made this work more complete. We have incorporated the newly added results in the revision. Please let us know if there is more to clarify.

---

> ### Author Response · Authors · 2022-11-16
> **Response to Reviewer yb7Z (1/2)**
>
> We thank Reviewer yb7Z for appreciating the novelty, clarity, and effectiveness of our work. Below, we address your main concerns.
>
> ---
>
> **Q1.** Add explanations on why a stronger attack hurts natural accuracy.
>
> **A1.** Thanks for pointing it out. Generally speaking, all features will become less robust under a stronger attack, and thus more (useful but non-robust) features will be discarded by adversarial training (because it only utilizes robust features). Consequently, there will be a smaller amount of usable features in the classifier, which hurts natural accuracy. We have added this explanation in **Section 3.4** in the revision.
>
> ---
>
> **Q2.** The definition of Eq. (3) seems confusing.
>
> > The trainer T is directly applied to update the model, i.e., changing the learned representations, thus, introducing the trainer T, given the representation, seems meaningless, but the trainer T is important for the rest of the work. Hence, the authors should clarify the point. Similarly, the utilized F_f^T is also confusing.
> >
>
> **A2.** Here, the trainer $\mathcal{T}$ refers to a general **training configuration** (lr, scheduler, weight decay, etc.), and $\mathcal{F}^\mathcal{T}$ refers to the set of attainable models under the current training configuration $\mathcal{T}$. In deep learning, the training configuration has a large impact on the learned models: we can get very different models by training the same architecture with different configurations. Therefore, as the effect of LR decay is a major concern of our work, we explicitly highlight the effect of the trainer $\mathcal{T}$ on the set of attainable models by denoting $\mathcal{F}^\mathcal{T}$. We have elaborated it in **Section 2** of the revision.
>
> ---
>
> **Q3.** The claimed theory seems to be a hypothesis, as merely experimental analysis is given in the current version.
>
> **A3.** Indeed, as mentioned in Sec 2.3, our explanation on robust overfitting is hypothetical, while **we provide extensive real-world experiments to verify this hypothesis**. In fact, the robust feature theory of Ilyas et al is also a hypothesis (mentioned in their Sec 3) and supported by experiments, and **we motivate our theory by some phenomena inexplicable in theirs**. Considering the fact that the adversarial training process of neural networks is extremely hard for analytical study, we can still explore the underlying mechanism from a phenomenological approach. Like the big bang theory and the hypothesis of continental drift, we believe that hypothesis is also an effective way to approach the truth **as long as it is verifiable**.
>
> ---
>
> **Q4.** A a straightforward approach to prevent fast LR decay can be that the trainer decreases the learning rate progressively from the previous epoch decaying the learning rate by 0.1.
>
> **A4.** Following your suggestion, we experiment with **Cosine and Linear LR schedule** on CIFAR-10 based on PreActResNet-18 architecture when the perturbation norm is 8/255. To be specific, the LR gradually decreases to 0.01 at epoch 150 and 0.001 at epoch 200 with a cosine / linear schedule. Here, we report the AA robustness of weight average (WA) model as it has higher robustness.
>
> | LR schedule | Loss | best | last | overfitting (best-last) |
> | --- | --- | --- | --- | --- |
> | piecewise | AT | 49.92 | 43.82 | 6.10 |
> |  | BoAT (ours) | 50.59 | 49.07 | 1.52 |
> | cosine | AT | 50.79 | 43.87 | 6.92 |
> |  | BoAT (ours) | 51.01 | 49.76 | 1.25 |
> | linear | AT | 50.83 | 44.37 | 6.46 |
> |  | BoAT (ours) | 51.17 | 49.63 | 1.54 |
>
> It can be seen that **a progressive LR decay (linear or cosine) cannot mitigate the robust overfitting (RO) problem**, as the overfitting rarely decreases: 6.1 (piecewise), 6.9 (cosine), and 6.5 (linear). Instead, we can see that our BoAT loss can effectively **mitigate RO for all kinds of LR schedules, by narrowing the overfitting gap from 6% to 1.5%** (in this setup). This shows that BoAT is robust and generalizable across different LR schedules. We added this experiment to Appendix C.3.
>
> ---
>
> **Q5.** The section on explanation seems weak, I suggest employing more references to convince the claim, the related perspective is the spurious feature perspective, such as CausalADV and group DRO.
>
> **A5**. Thanks for bringing up these related works. CausalADV provides a causal explanation of non-robust features as spurious features between label Y and style S, which naturally fits our explanations: when spurious features are falsely memorized (in the OOD sense as in group DRO), it would also induce significant performance degradation. In this regard, our proposed method (BoAT) can be seen as explicit regularizations for suppressing spurious features. We have added this discussion in **Section 2.3.**

---

> > ### Comment · Reviewer_p7yt · 2022-11-27
> > **A further question**
> >
> > If $\mathcal{T}$ stands for training configuration, shouldn't the $\mathcal{T}$ be included in Equation (2)? In the original paper, they also considered these configurations including LR decay, weight decay, and etc. (See Table 3 in https://arxiv.org/pdf/1905.02175.pdf)?

---

> > > ### Author Response · Authors · 2022-12-02
> > > **Further comparison on robustness definition**
> > >
> > > Thanks for your response. To answer this question, we need to realize that there are actually **two separate notations of "robustness" considered in Ilyas et al (2019): feature robustness and model robustness**. In Eq.2 (our paper), we are **quoting** their definition of *feature robustness* in Eq. 1 & 2 in their paper (page 3), where a feature is a function mapping $f:\mathcal{X}\to\mathbb{R}$, and the robustness of $f$ is defined as the prediction accuracy of **$f$ itself** under perturbations. We can clearly see that their definition is **independent of any specific models used to learn this feature.** In other words, they believe that the robustness of a feature is a model-agnostic **data property**.
> > >
> > > Then, in the “Classification” paragraph, they define a **model** as a weighted combination of these static data features. Thus, different models with different **weights** can have different robust accuracies (which we dubbed as **model robustness**). From this comparison, we can see that in their framework, the difference in model robustness is NOT caused by the difference in feature robustness, but only by different model capacities or training configurations. Therefore, to answer your problems, **their definition of feature robustness is indeed static and not involves model configurations**.
> > >
> > > In our work, a motivation for our dynamic view of **feature robustness** is the following contradiction of Ilyas et al. (2019): if feature robustness is simply a data property and independent of the model (as their definition suggests), then **learning these robust features from data with adversarial training should be as normal as standard training**. In particular, as in standard training of deep models, a change of model trainer (e.g., LR decay) should NOT induce a severe overfitting phenomenon, which is contradictory to the practice. Therefore, we believe that their static view of feature robustness cannot explain the very unusual robust overfitting behavior in adversarial training. To address this problem, in this work, we further take the trainer into the definition of feature robustness. We believe the robustness of a feature is not solely a data property, but instead depends on the dynamic balance between the trainer and the attacker, and we validate this perspective on real-world data.
> > >
> > > Hope our clarification above could address your concerns. Please let us know if you have additional concerns.

---

### Official Review · Reviewer_zGeX · 2022-10-27

**Confidence:** 3
**Correctness:** 2
**Technical Novelty And Significance:** 2
**Empirical Novelty And Significance:** 2
**Recommendation:** 5

**Clarity, Quality, Novelty And Reproducibility:**

I think the paper can be drastically improved.

Figure out what the key contribution is and really support that well. I am not sure the conceptual framework of robust features from Ilyas et al. contributed to the conceptual understanding since you ended up changing their definitions anyway.

The writing is tentative. For instance: "Therefore, due to the change of relative strength between A and T , these non-robust features could become robust after LR decay."

The stated findings are not particularly enlightening. For example:

"The discrepancy in feature robustness explains why the rise of training robustness does not consistently lead to higher test robustness, leading to a large robust generalization gap.

It is not clear that one thing explains the other.  What about robust features? Is their overfitting ruled out?
And the above statement begs the question as to why non-robust features cause overfitting.

Performance Experiments:

How do your findings compare with Yu et al.  in ICML 2022 (Understanding Robust Overfitting of Adversarial Training and Beyond)? They proposed a method called minimum-loss constrained adversarial training (MLCAT).  And what about efficiency of training?

**Strength And Weaknesses:**


Strengths:

I am still trying to figure that out. I wish the paper was clearer in writing. It seems the authors are trying to draw a connection between feature robustness (using their definition, not Ilyas et al.), learning rate decay,  and overfitting. And why the authors think that standard training doesn't run into overfitting but adversarial training does at a fundamental level.

In the end, it is not clear if BoAT is a better algorithm over prior methods for adversarial training.

Weaknesses:

English writing is poor, making the paper difficult to follow. The abstract could be better written to begin with.

Limitations on page 3: I didn't understand your argument. I didn't find in Ilyas a discussion or claim regarding overfitting.  Furthermore,  during model training, isn't the classifier changing? And also isn't f changing, if if is part of the model? If it is some abstract function f independent of the trained model, then why you do believe that robustness of f is changing during model training?

Furthermore, I don't see an attacker A in definition (2).

I don't agree with the characterization of  Ilyas et al. 2019 that they only consider attacker A and fail to view adversarial training as a dynamic min-max game. For instance, in Robust Training para on page 4 of Ilyas et al., they explicitly discuss adversarial training a min-max optimization problem.

Furthermore, the definition I Alias et al. considers the adversarial space, but does not explicitly mention an attacker strategy. I am still trying to understand your criticism of Ilyas et al. 2019.






**Summary Of The Paper:**

The paper cites the feature robustness framework of Ilyas et al. 2019 as its starting point. It claims that framework fails to explain overfitting in adversarial training (Note: that paper wasn't claiming to explain overfitting). The paper then proposes a narrower definition of feature robustness (where an attacker and a trainer are given). They claim that it explains overfitting.

The paper, I think, mischaracterizes Ilyas et al. 2019 definition of feature robustness when describing Limitations.

The writing is not particularly clear on why the change in definition explains overfitting, which was the original motivation for the paper. The paper acknowledges past work that robust overfitting can be largely "surpassed" (suppressed?) by enforced a smoother landscape with additional regularizations. In the end, it seems the proposed schemed BoAT also ends up doing something similar (Section 3.1: "In order to suppress robust overfitting of the model trainer T after LR decays, we need to put further regularization on its local fitting ability, e.g., by enforcing better landscape flatness.").

So, I am still trying to figure out the key contribution of the paper and new teachings. It could be that the theoretical arguments need improvement to explain overfitting.

The BoAT training method is a potential contribution since the claim is that it eliminates overfitting even over 500 epochs. But, the authors should compare that with other approaches such as simply picking an earlier saved model (prior to overfitting) and prior techniques that introduce additional regularizations to smooth the landscape (many of which they cite in the 2nd para of the Intro).

The other issue is efficiency of BoAT training. How long does the training take compared to existing techniques, e.g.,  early stop? I am trying to figure if BoAT is proposed as a practical technique that practitioners should use or is simply a tool to help explain overfitting.

The Intro states:

"Specifically, we hypothesize that due to the strong local fitting ability endowed by smaller LR, the model learns false mapping of non-robust features contained in adversarial examples, which opens shortcuts for the test-time attacker and induces large test robust error. "

But, why doesn't standard training also learn false mappings on non-robust features and also cause overfitting?

In the end, the paper started with an interesting promise of explaining the difference in overfitting between standard training and adversarial training. But, that difference in behavior still remains unclear after reading the paper.


**Summary Of The Review:**

The paper needs to be better written with a  more careful discussion of related work to better position the paper. The paper is also missing some related work that also attempts to explain overfitting and propose techniques to prevent overfitting.

Additional comments for the authors after the rebuttal:

I thank the reviewers for detailed explanations. I think the algorithm is a good contribution. But, I still feel that the overall pitch in the Intro and Section 2 should be changed in a future version of the paper. I am not sure Ilyas et al. is the correct baseline to criticize, since it was not directed at providing an explanation on robust overfitting. Other papers that attempt to explain that in other ways would be a better starting point.  In case you use Ilyas et al., perhaps a better pitch would be to say that even if a feature is fundamentally robust,  it can be perceived as non-robust by a given model because of inability to find the correct decision boundary for that feature -- even if one exists.  That way, perhaps Ilyas et al. are not necessarily wrong -- static and dynamic views become more in harmony. I don't think the paper proves that their view was necessarily wrong, but the paper suggests so, which I think is an issue with the current thrust.

Another area that the paper could  do a better job in  would be to better explain why normal training does not overfit, but adversarial does. Are there situations where normal training would also overfit, based on the understanding provided by the work? Even normal data should have some non-robust features, no -- or is that fundamentally missing in normal datasets?  More in-depth discussion would be useful.

Overall, I am not changing my evaluation, but  I encourage authors to rethink the overall pitch for presenting the Boat algorithm and add more depth for an improved paper in the future.

---

> ### Author Response · Authors · 2022-11-16
> **Response to Reviewer zGeX (2/2)**
>
>
> **Q3.** How long does the training take compared to existing techniques, e.g., early stop? I am trying to figure if BoAT is proposed as a practical technique that practitioners should use or is simply a tool to help explain overfitting.
>
> **A3.** We note that early stopping is a widely adopted technique in adversarial training due to the robust overfitting phenomenon, and it is **orthogonal to our proposed BoAT method**. Therefore, in Table 1, we report both best-epoch (with early stop) and last-epoch (without early stop) results with BoAT. We can see that our method obtains **clear improvements in robustness with or without early stopping method**. Meanwhile, as revealed in **A2** above**,** our method is also very computationally efficient, making it very suitable for practical use.
>
> ---
>
> **Q4.** The stated findings are not particularly enlightening. For example:
>
> > *"The discrepancy in feature robustness **explains why** the rise of training robustness does not consistently lead to higher test robustness, leading to a large robust generalization gap.*
> >
>
> It is not clear that one thing explains the other. What about robust features? Is their overfitting ruled out? And the above statement begs the question as to why non-robust features cause overfitting.
>
> **A4.** Thanks for pointing out the unclarity here. A better expression is that “the discrepancy in feature robustness **shows that** the rise of training robustness does not consistently lead to higher test robustness …” The results are only designed to only **show the existence of robust generalization gap**. We have modified it in the revision. As for how non-robust features cause overfitting, we leave it to **Sec 2.2** for a thorough discussion with additional evidences.
>
> ---
>
> **Q5.** Comparison to MLCAT by Yu et al. in ICML 2022 (Understanding Robust Overfitting of Adversarial Training and Beyond)? And what about efficiency of training?
>
> **A5. Comparison of Methodologies:**
>
> - Yu et al. (2022) [1] analyze robust overfitting by investigating **the roles of easy and hard samples** and propose to regularize samples of small loss values (typically using non-robust features)
> - Nevertheless, their discussions cannot fully explain how robust overfitting rises from easy samples. In comparison, we provide a generic explanation for the rise of robust overfitting from **a feature dynamics perspective**, and we verify this understanding through extensive experiments. Our proposed BoAT is also designed from a bootstrapping perspective, which is different from theirs.
> - At last, our understanding can also help understand how Yu et al’s method works: by regularizing small-loss data, they enhance robust features and diminish non-robust features, which helps alleviate robust overfitting.
>
> **Comparison of Performance and Efficiency.** Below, we further compare the empirical performance between their proposed MLCAT and our BoAT following your suggestions.
>
> #### *a) Performance on CIFAR-10 with PreActResNet-18*
>
> | Method | best | last | diff |
> | --- | --- | --- | --- |
> | MLCAT | 50.70 | 50.32 | 0.38 |
> | BoAT (ours) | 51.39 | 51.36 | 0.03 |
>
> #### b) *Performance* on CIFAR-100 with PreActResNet-18
>
> | Method | best | last | diff |
> | --- | --- | --- | --- |
> | MLCAT | 25.66 | 25.28 | 0.38 |
> | BoAT (ours) | 27.59 | 27.52 | 0.07 |
>
> *#### c) Training time on CIFAR-10 with PreActResNet-18*
>
> | Method | Time per epoch |
> | --- | --- |
> | MLCAT | 353.3s |
> | BoAT (ours) | 136.2s |
>
> Tables a and b above show that 1) our BoAT method is consistently superior (>1% AA improvements) to MLCAT across different datasets, and 2) BoAT has weaker robust overfitting than MLCAT. Table c further shows that 3) BoAT is much more training efficient by **requiring only 1/3 training time of MLCAT**. Therefore, BoAT is both ***more effective, more efficient, and has less overfitting*** than MLCAT [1].
>
> Following your suggestion, we have discussed MLCAT to Related Work (Section 5), and added this comparison in Appendix C.7.
>
> Reference:
>
> [1] Chaojian Yu, Bo Han, Li Shen, Jun Yu, Chen Gong, Mingming Gong, and Tongliang Liu. Understanding robust overfitting of adversarial training and beyond. In ICML, 2022.
>
> ---
>
> Thanks for your careful reading and constructive comments. We have revised our writing according to your suggestions, and elaborated our key differences to prior works. Please let us know if you have additional concerns. We are looking forward to hearing from you.

---

> ### Author Response · Authors · 2022-11-16
> **Response to Reviewer zGeX (1/2)**
>
> We thank Reviewer zGeX for your time and efforts on reviewing this work. Below, we address your main concerns of this work.
>
> ---
>
> Q1. Concerns on the limitations of Ilyas et al.
>
> A1. We address your concerns point by point.
>
> > **1) what is wrong in Ilyas’ theory? I didn't find in Ilyas a discussion or claim regarding overfitting.**
> >
>
> The point that we tried to make here is that Ilyas’s theory **cannot explain** the rise of robust overfitting. The key point of their theory is that **robust and non-robust features are intrinsic data properties**. Therefore, model robustness reflects the robustness of utilized data features. However, our point is that while **data features remain the same across training, the change of model trainer, e.g., LR decay, indeed has a dramatic influence on model robustness**. Ilyas’ theory cannot explain this. Therefore, robustness is certainly **not (only) a data property**, but crucially depends on the interplay between trainer and attacker, which is reflected in our new definition of robustness. As a result of the dynamic interplay, it will change if the two game players change.
>
> > **2) Is f changing? if it is independent of model,  then why its robustness is changing?**
> >
>
> In both Ilyas’ and our definition, f refers to a data feature (independent of model). Thus, the feature itself is not changing. Instead, the main difference is that **Ilyas thinks “the robustness of a feature” is static, while we think it is dependent on model configurations**.
>
> For example, **the same data feature can be robust under a linear model, while non-robust under an MLP**. Consider a two-dimensional mixture of Gaussian data of two classes, where data from class 1 follow $N((1,0), I)$, and data from class -1  follow $N((-1,0),I)$.  We can else adopt a linear classifier and an MLP classifier to fit this data, and they will learn different decision boundaries. Then, when utilizing the x-axis feature of the data for attack (perturbing along the direction (1,0)), the attacker will achieve different success rates under two models. Therefore, the same x-axis data feature could have very different robustness under two different models. Similarly, a feature can be **robust under a weak attacker (smaller epsilon, less steps), and non-robust against a strong attacker.**
>
> To summarize, **a feature f does not change, but its robustness may change** along training due to the dynamic change of the model trainer and the attacker. In practice, in **Fig1c**, we have also verified **the same inputs indeed have very different robustness under different training configurations.**
>
> > **3) I don't see an attacker A in definition (2).**
> >
>
> The strength of the attacker is reflected in its attacker budget $\Delta$ in Eq. 2.
>
> > **4) Ilyas et al. 2019 also regard AT as a dynamic min-max game. What is the key contribution of this work?**
> >
>
> As elaborated above, our key point is that Ilyas et al think that the **robustness of a feature** is a data property, while we find that ***robustness of a feature* depends on the dynamics of min-max game**. Although Ilyas mentioned that AT is a dynamic game, they **only consider its equilibrium, and believe the equilibrium classifier only extracts features based on their intrinsic robustness.** However, ****this is not true. They do not realize that the equilibrium even might not exist, such as, when the model keeps overfitting. We also revealed that LR and attacker changes have a dramatic influence on feature robustness. A key contribution compared to Ilyas et al is that we are the first to take the min-max nature of AT into **the definition of robustness**.
>
> ---
>
> **Q2.** Training efficiency of BoAT.
>
> **A2.** The proposed BoAT method is computationally efficient as it **requires only 1 extra forward propagation of the WA model per iteration compared with WA**, whose computational cost is almost neglectable since the time consumption is mainly on generating adversarial examples (10 forward propagations and adversarial noise updates for PGD-10). For a fair comparison, all the compared methods are integrated into a universal training framework and each test runs on a single NVIDIA GeForce RTX 3090 GPU.
>
> | Method |  Training Time per Epoch (s) |
> | --- | --- |
> | PGD-AT | 131.6 |
> | WA | 132.1 |
> | KD+SWA | 289.8 |
> | AWP | 142.8 |
> | MLCAT | 353.3 |
> | BoAT (ours) | 136.2 |
>
> From the table above, we can see that BoAT indeed **requires nearly no extra computational cost** compared with vanilla PGD-AT (136.2s v.s. 131.6s per epoch), implying that it is an efficient training method in practice.  We have added this in Appendix C.6.

---

> ### Comment · Reviewer_p7yt · 2022-11-27
> **Thanks for answering my questions**
>
> I want to thank the authors for the detailed response. I am now convinced that BoAT is efficient enough compared with competitors. However, there are two major claims that I am not convinced of:
>
> 1) The connection between LR decay and the dynamic feature
>
> - The experiments in Figure 1 seem to me that without LR decay, the model used in the experiment under-fits a lot (in other words, trained improperly). Even without LR decay, one with a large capacity model and a properly tuned fixed LR, that person should be able to fit the training robustness to 100%. I did not see any connection specifically between LR decay and the dynamic features. In addition, the concept of dynamic features is still very vague. The authors mentioned using robustness as a proxy to measure the dynamics of features, but is there any evidence showing that this is a good measurement? Is there any concrete example of how the features changed so that it caused overfitting?
> - The authors do show LR decay has a confusion matrix that is more symmetric, and LR decay also has better test robustness. However, this does not mean that because the confusion matrix becomes more symmetric, the model becomes more robust (a random classifier has a perfectly symmetric but does not have a good robust test accuracy).
> - How does the LR decay differ from another regularization method? It is already well-known that early stopping and weight decay (https://arxiv.org/pdf/2002.11569.pdf) can reduce robust overfitting. Re-running all experiments in Figure 1 and comparing between with and without weight decay, one can achieve a similar result.
>
> There are also many places where terms are misused, which makes the entire paper hard to understand.
> - The paper uses better robust test accuracy to imply less robust overfitting. This is not correct, as they are two different concepts.
> - The author uses Figure 1(d) to claim the memorization of the non-robust features. However, the figure only shows the relative changes in test robustness. It does not show any measurements on memorization. From the experiment alone, we are unable to tell whether the drop in robustness is caused by memorization or other factors that are not measured. In other words, being robust to some features does not imply memorizing those features (there are separate measurements for memorization https://arxiv.org/pdf/2008.03703.pdf).
>
> 2) The claim of SOTA performance
>
> - To claim SOTA performance, using the testing dataset for early stopping is NOT acceptable. Even the result may be similar to using the validation set for early stopping. The numbers reported should be for the model that uses a validation set for early stopping.
> - Within Tables 1 and 2, it is unclear how the parameters for other competitors are tuned. Thus, I could not tell whether it is a fair comparison (if one model is tuned extensively while the other one is not, this is not a fair comparison). When comparing with the external leaderboard (i.e., https://robustbench.github.io/), it does not seem to me that BoAT is the best-performing one, at least for CIFAR10 and CIFAR100.
>
> Based on these reasons, I would like to keep my current score.

---

> > ### Author Response · Authors · 2022-12-02
> > **Further Response to Reviewer  p7yt (3/3)**
> >
> > **Q7.** To claim SOTA performance, using the testing dataset for early stopping is NOT acceptable. Even the result may be similar to using the validation set for early stopping. The numbers reported should be for the model that uses a validation set for early stopping.
> >
> > **A7.** As explained in the previous reply, for a consistent comparison, we follow the common test-based early stopping adopted in the baseline methods including TRADES, TE, and AWP. Meanwhile, we also think your advice is valuable and it would be better if all these methods would prefer validation-based early stopping. Following your advice, we **further re-produce all our results using validation-based early stopping** (no further tuning) and report the results and comparisons below.
> >
> > Table a. ***Results under validation-based early stopping with a PreActResNet-18 backbone***
> > | Dataset | Method | best natural | last natural | best AA | last AA | ↓ diff (best-last AA) |
> > | --- | --- | --- | --- | --- | --- | --- |
> > | CIFAR-10 | PGD-AT | 81.61 | 84.67 | 47.51 | 42.31 | 5.2 |
> > |  | BoAT | 81.86 | 81.91 | 51.13 (+3.62) | 51.22 (+8.91) | -0.09 |
> > | CIFAR-100 | PGD-AT | 55.94 | 56.39 | 24.84 | 19.80 | 5.04 |
> > |  | BoAT | 56.13 | 56.79 | 27.60 (+2.76) | 27.29 (+7.49) | 0.31 |
> > | Tiny-ImageNet | PGD-AT | 45.29 | 48.70 | 17.29 | 14.19 | 3.10 |
> > |  | BoAT | 48.28 | 47.92 | 20.54 (+3.25) | 19.79 (+5.60) | 0.75 |
> >
> > (see WideResNet-34 results in the new response below)
> >
> > We can see that under validation-based early stopping, on all benchmark datasets (CIFAR-10, CIFAR-100, Tiny-ImageNet), our method still consistently outperforms the baseline significantly. Even comparing to the test-based results of other methods in Tab 1 & 2, our method also enjoys much better robustness with nearly no robust overfitting. For a fair comparison, **we are now reproducing all baseline methods under the validation-based setting, and will report these results in the revision**. We believe that our benchmark under this setting would encourage a more fair comparison in the community.
> >
> > ---
> >
> > **Q8.** Within Tables 1 and 2, it is unclear how the parameters for other competitors are tuned. Thus, I could not tell whether it is a fair comparison (if one model is tuned extensively while the other one is not, this is not a fair comparison).
> >
> > **A8.** Many methods in Tables 1 and 2 require several parameters to tune. Among the competitive ones, **AWP introduces 4 tunable parameters**, such as, 1) the step number, 2) step size, and 3) perturbation budget for its additional weight perturbation step, and 4) the alternate iteration number between the two perturbations; **KD+SWA introduces 3 tunable parameters**, 1) distillation parameter, 2) coefficient for ST teacher, and 3) coefficient for AT teacher. As far as we could see, these parameters have great influence on their performance, and their choice is extensively selected. In comparison, **BoAT only introduces 2 parameters**, the loss coefficient $\lambda$ and the decay factor $d$, and our method is relatively robust to small perturbations of these parameters. Therefore, we believe that our method is easy to use with strong performance. We will fill in more details of their comparisons in the revision.
> >
> > ---
> >
> > **Q9.** When comparing with the external leaderboard (i.e., [https://robustbench.github.io/](https://robustbench.github.io/)), it does not seem to me that BoAT is the best-performing one, at least for CIFAR10 and CIFAR100.
> >
> > **A9.** In order to attain SoTA robustness, many methods on RobustBench require larger backbone networks (WideResNet-70), module replacement (e.g., ReLU → GELU), and a lot of strong augmentations and additional data (e.g., 100M synthetic datasets or extra datasets). These are well-known techniques that help robustness but require a lot more computation. Our proposed BoAT is orthogonal to these techniques, and we mainly focus on the learning objective itself. Therefore, as done in many works, we compare under the same vanilla network and the same vanilla data augmentation. Under this scenario, for example, searching ResNet-18 in RobustBench, we can find that the best performing method to date (without using additional augmentations and synthetic data) is [Towards Achieving Adversarial Robustness Beyond Perceptual Limits](https://openreview.net/forum?id=SHB_znlW5G7) with 51.06% AA. In comparison, our BoAT attains 51.36% AA (final-epoch, no early stopping), making it a SoTA method under this vanilla setting, showing the effectiveness of BoAT.  Following your suggestions, we will clarify this point and tune down the SoTA claims to be more specific in the revision.
> >
> > ---
> >
> > Thanks for your valuable and constructive comments. Hope our explanations and newly reproduced results could address your concerns. Please let us know if there is more to clarify. We are looking forward to your further reply.

---

> > > ### Author Response · Authors · 2022-12-04
> > > **Further Response to Reviewer p7yt (WideResNet results added)**
> > >
> > > To complement the results of **A7** above, we further evaluate our method against PGD-AT on a large backbone network, WideResNet-34, using validation-based early stopping.
> > >
> > > Table b. ***Results under validation-based early stopping with a WideResNet-34 backbone on CIFAR-10***
> > >
> > > | Method | best natural | last natural | best AA | last AA | ↓ diff (best-last AA) |
> > > | --- | --- | --- | --- | --- | --- |
> > > | PGD-AT | 86.38 | 87.04 | 51.94 | 45.87 | 6.07 |
> > > | BoAT | 85.25 | 85.52 | 54.78 (+2.84) | 54.80 (+8.93) | -0.02 |
> > >
> > > We can see that under validation-based early stopping, our method indeed has **no overfitting** (final AA > best AA), and meanwhile, obtains much better best and final AA robustness than PGD-AT. Meanwhile, it performs significantly better than other baselines in Table 1. It shows our proposed BoAT indeed alleviates the robust overfitting problem of AT.
> > >
> > > Hope our explanations and the new results above could address your concerns. Please let us know if there is more to clarify.

---

> > > ### Author Response · Authors · 2022-12-06
> > > **All benchmark results reproduced under validation-based early stopping**
> > >
> > >
> > > Dear Reviewer p7yt,
> > >
> > > We have now reproduced the benchmark results of all baseline methods as well as our method (Table 1 & 2) under the validation-based early stopping, and added them below for reference. We can see that they are **close and consistent with the original results**. In particular, our method is still **consistently and significantly better than others** with a neglectable degree of robust overfitting. We will update these results in the revision.
> > >
> > > Hope you find this result satisfactory. Please let us know if there is more to clarify.
> > >
> > >
> > > Best,
> > >
> > > Authors
> > >
> > > ---
> > >
> > > **Table 1 (reproduced)**. Best AA robustness among all methods is highlighted.
> > >
> > > | CIFAR-10 | Res18 |  |  |  |  |  |  | Wide34 |  |  |  |  |  |
> > > | --- | --- | --- | --- | --- | --- | --- | --- | --- | --- | --- | --- | --- | --- |
> > > |  | Natural |  | PGD-20 |  | AA |  |  | Natural |  | PGD-20 |  | AA |  |
> > > | method | best | last | best | last | best | last |  | best | last | best | last | best | last |
> > > | PGD-AT | 81.61 | 84.67 | 52.51 | 46.20 | 47.51 | 42.31 |  | 86.38 | 87.04 | 56.62 | 48.54 | 51.94 | 45.87 |
> > > | TRADES | 80.45 | 82.89 | 52.32 | 49.39 | 48.09 | 46.40 |  | 84.23 | 84.89 | 56.02 | 47.38 | 52.76 | 45.62 |
> > > | WA | 83.50 | 84.94 | 55.05 | 47.60 | 49.89 | 43.83 |  | 87.66 | 87.12 | 56.62 | 49.12 | 52.65 | 46.34 |
> > > | KD+SWA | 84.06 | 84.48 | 53.70 | 53.68 | 49.82 | 49.37 |  | 87.45 | 88.21 | 56.29 | 55.76 | 53.59 | 53.55 |
> > > | PGD-AT+TE | 82.04 | 82.59 | 54.98 | 53.54 | 50.12 | 49.09 |  | 85.97 | 85.77 | 56.42 | 53.40 | 52.88 | 50.56 |
> > > | AWP | 81.11 | 80.62 | 55.60 | 55.03 | 50.09 | 49.85 |  | 85.63 | 85.61 | 58.95 | 59.05 | 53.32 | 53.38 |
> > > | BoAT | 81.86 | 81.91 | 56.36 | 56.12 | **51.13** | **51.22** |  | 85.25 | 85.52 | 59.59 | 59.29 | 54.78 | 54.80 |
> > > | WA+CutMix | 80.24 | 80.30 | 56.02 | 55.95 | 49.67 | 49.59 |  | 85.08 | 87.95 | 60.41 | 59.36 | 55.11 | 53.60 |
> > > | BoAT+CutMix | 79.02 | 78.95 | 56.15 | 56.16 | 50.18 | 50.22 |  | 86.28 | 87.01 | 61.33 | 61.24 | **55.75** | **55.72** |
> > >
> > > **Table 2 (reproduced)**. Best AA robustness among all methods is highlighted.
> > >
> > > |  | CIFAR-100 |  |  |  |  |  |  | Tiny-ImageNet |  |  |  |  |  |
> > > | --- | --- | --- | --- | --- | --- | --- | --- | --- | --- | --- | --- | --- | --- |
> > > |  | Natural |  | PGD-20 |  | AA |  |  | Natural |  | PGD-20 |  | AA |  |
> > > | method | best | last | best | last | best | last |    | best | last | best | last | best | last |
> > > | PGD-AT | 55.94 | 56.39 | 29.74 | 22.47 | 24.84 | 19.80 |  | 45.29 | 48.70 | 21.86 | 17.54 | 17.29 | 14.19 |
> > > | TRADES | 55.04 | 57.09 | 29.17 | 26.36 | 24.21 | 23.06 |  | 48.32 | 47.59 | 22.25 | 21.14 | 16.55 | 15.99 |
> > > | WA | 57.26 | 58.40 | 30.35 | 23.52 | 25.83 | 20.51 |  | 49.56 | 49.35 | 24.24 | 19.30 | 19.47 | 15.56 |
> > > | KD+SWA | 57.17 | 58.23 | 29.50 | 29.33 | 25.66 | 25.65 |  | 50.28 | 50.67 | 24.37 | 24.30 | 19.46 | 19.51 |
> > > | PGD-AT+TE | 56.41 | 57.26 | 30.90 | 29.07 | 25.84 | 24.99 |  | 46.71 | 50.50 | 22.76 | 19.26 | 18.02 | 16.13 |
> > > | AWP | 54.10 | 54.78 | 30.47 | 30.15 | 25.16 | 25.01 |  | 43.54 | 43.33 | 23.75 | 23.55 | 18.12 | 18.07 |
> > > | BoAT | 56.13 | 56.79 | 32.52 | 32.02 | **27.60** | **27.29** |  | 48.28 | 47.92 | 24.93 | 24.14 | **20.54** | **19.79** |
> > > | WA+CutMix | 56.73 | 57.17 | 31.70 | 31.71 | 26.43 | 26.08 |  | 46.48 | 46.48 | 24.80 | 24.72 | 18.87 | 18.87 |
> > > | BoAT+CutMix | 56.05 | 56.02 | 32.71 | 32.57 | 27.08 | 27.20 |  | 46.47 | 46.62 | 25.28 | 25.28 | 19.40 | 19.35 |

---

> > ### Author Response · Authors · 2022-12-02
> > **Further Response to Reviewer p7yt (2/3)**
> >
> > **Q3.** The authors do show LR decay has a confusion matrix that is more symmetric, and LR decay also has better test robustness. However, this does not mean that because the confusion matrix becomes more symmetric, the model becomes more robust (a random classifier has a perfectly symmetric but does not have a good robust test accuracy).
> >
> > **A3.** We are afraid that there are some misunderstandings here. In this experiment, we are NOT drawing a relationship between confusion symmetry and better test robustness, but instead, **a relationship between confusion symmetry and robust overfitting**, more specifically, showing **how severe robust overfitting leads to confusion symmetry** (explained and verified in Sec 2.3, 4th paragraph). In Figure 4c, we show that our BoAT indeed alleviates this bilateral correlation by alleviating robust overfitting.
> >
> > ---
> >
> > **Q4.** How does the LR decay differ from another regularization method? It is already well-known that early stopping and weight decay ([https://arxiv.org/pdf/2002.11569.pdf](https://arxiv.org/pdf/2002.11569.pdf)) can reduce robust overfitting. Re-running all experiments in Figure 1 and comparing between with and without weight decay, one can achieve a similar result.
> >
> > **A4.** Indeed, there are many factors that affect RO, including weight decay. Nevertheless, as explained above, under the best training configuration explored by Rice et al. (2020), RO still exists, and **LR decay is a key step for triggering RO during training**. Therefore, we believe that LR decay is a representative example to study how training configuration affects robustness. As you pointed out, our analysis could also be applied to other factors, such as, weight decay, and may obtain similar results. **Since the main focus of this work is to establish a dynamic view of feature robustness, we believe that the LR decay example has clearly illustrated and verified this point.** We will keep exploring the effect of other factors in the future.
> >
> > ---
> >
> > **Q5.** The paper uses better robust test accuracy to imply less robust overfitting. This is not correct, as they are two different concepts.
> >
> > **A5.** We are afraid there are some misunderstandings here. We do realize the two concepts are different, which is why we report both best-epoch and last-epoch robustness in Tab. 1 & 2 to present the robust overfitting degree (best - last). In our analysis in Sec 4.1, we also compare the gap between best-epoch and last-epoch robustness to elaborate how our method alleviates overfitting.
> >
> > In the meanwhile, as you mentioned, we also pay attention to achieving better robust test accuracy. This is because solely avoiding RO is a trivial task. For example, as shown in Fig 1b, we can avoid RO by simply dropping LR decay. However, it will also lead to much worse model robustness (about ↓5%). This shows that **one can always easily avoid RO in AT if he does not care about degrading robust accuracy**. Therefore, the challenging thing here (and the goal of our method) is to alleviate RO while maintaining (or better, improving) the best robust accuracy. This is why we focus on both RO and best robust accuracy. As shown in Table 1, our BoAT attains much better robust accuracy (51.36% final) with very slight RO (51.39% best and 51.36% final), completing the two goals at a time.
> >
> > ---
> >
> > **Q6.** Figure 1(d) does not show any measurements on memorization.
> >
> > **A6.** We are afraid that there are also some misunderstandings here. The memorization of non-robust features is NOT justified by **Figure 1d**. Instead, it is evidenced by **Figure 1c** and discussed in Sec 2.2, 3rd paragraph (before discussing Figure 1d). Specifically, we show that the LR decay introduces **a significant increase in training robustness of non-robust features**, but this increase does not lead to an improvement in their test robustness, leading to a large robust generalization gap. Accordingly, we believe the non-robust features are simply “memorized” but not generalizable during this process. In Sec 2.3 and Figure 1d, we further show how these “memorized” features hurt test robustness. We will revise our writing to be more clear on this logic.
> >
> > Meanwhile, for a formal study of the memorization ability, we indeed need more evidence, e.g., on the random label setting, as done in the ICLR 2022 paper [Exploring Memorization in Adversarial Training]([https://openreview.net/forum?id=7gE9V9GBZaI](https://openreview.net/forum?id=7gE9V9GBZaI)). In fact, they also observed a close relationship between memorization and RO, while our work could serve as an explanation to it. For better rigorism, we will reword the “memorized” features as “fitted” features in the revision as you suggested.

---

> > ### Author Response · Authors · 2022-12-02
> > **Further Response to Reviewer p7yt (1/3)**
> >
> > Thanks for your response and for appreciating the efficiency of BoAT. We will further address your remaining concerns below.
> >
> > ---
> >
> > **Q1.** The connection between LR decay and the dynamic feature.
> >
> > **A1.** Thanks for your careful reading. Though, before diving into your questions, please let us explain  “dynamic features” to avoid possible misunderstandings. In our theory, **features are NOT dynamic** (as they are independent of models, same as Ilyas et al), but **only the robustness of features is dynamic during training** (which Ilyas et al do not consider). Specifically, in our definition, robustness is NOT a data property as in Ilyas et al, but instead **dependent on the *dynamic interplay*** between model trainer and attacker during the training process as in Eq. 3. We will revise our paper to be more clear on this point.
> >
> > Based on these understandings, we further address the specific question you raised.
> >
> > > Even without LR decay, one with a large capacity model and a properly tuned fixed LR, that person should be able to fit the training robustness to 100%. I did not see any connection specifically between LR decay and the dynamic features.
> > >
> >
> > **Trainer factors that affect RO.** Indeed, as mentioned in Sec 2.2, there are many other factors in the trainer except LR, such as, optimizer, weight decay, and training epochs. According to our theory, the change of these factors also has an effect on feature robustness and robust overfitting. Here, we mainly take LR as **a representative example** to study the effect of breaking the balance between the model trainer and the attacker.
> >
> > **Why do we study LR decay?** Indeed, as you suggest, we could also fit the data by adopting a small LR right from the beginning, but empirically, **it typically leads to worse robustness**. Therefore, it is more of our interest to study RO under the common training recipe **yielding SOTA robustness**. In fact, the piecewise LR decay schedule is the *de facto* recipe for adversarial training and widely adopted in most influential AT papers, such as, Madry et al. (2018), TRADES (Zhang et al. 2019), and robust overfitting (Rice et al. 2020). In this common schedule, many have observed that **the first LR decay is the key trigger for the RO phenomenon:**
> >
> > > (quoted from *Rice et al. (2020)*) *This is shown, for instance, in Figure 1 for adversarial training on CIFAR-10, where the robust test error **dips immediately after the first learning rate decay, and only increases beyond this point**.*
> > >
> >
> > Our experiments in **Figure 7 (Appendix B.2**) also show that in this schedule, **decaying LR at intermediate epochs** (25, 50, 75, 100, 125, 150, 175) will always lead to robust overfitting. Because LR decay is such **a key operator to induce a dramatic RO in canonical AT,** we take it as **a representative example** to study **how robustness changes when the balance between the trainer** $\mathcal{T}$ **and the attacker** $\mathcal{A}$ **breaks.**
> >
> > **Application to other factors**. Similar to LR decay, the change of other factors during training may **also introduce an imbalance** and RO likewise. Therefore, **our analysis on the trainer-attacker imbalance** could also be applied to understanding other changing factors, e.g., weight decay.
> >
> > ---
> >
> > **Q2.** The authors mentioned using robustness as a proxy to measure the dynamics of features, but is there any evidence showing that this is a good measurement? Is there any concrete example of how the features changed so that it caused overfitting?
> >
> > **A2.** As explained above, in our theory, **it is the feature robustness (NOT the feature) that is dynamic along the training**. Therefore, **our goal is to analyze how feature robustness changes** along training, as we measured in Figure 1. The difficulty lies in the fact that as in Ilyas et al (2019), we cannot actually separate all features in an images for observing the robustness change of each feature. Inspired by Ilyas et al, we first distill non-robust features extracted from different learning stages, and conduct a controlled experiment on their effect on RO (Fig 1d) and investigate their properties by a series of experiments (Fig 2). These experiments indeed reveal the robustness properties that we care about and help verify our hypothesis.

---

> ### Author Response · Authors · 2022-12-04
> **Looking forward to your feedback**
>
> Dear Reviewer zGeX,
>
> Thanks for your careful reading of our paper. We have tried our best to elaborate the unclear points and revised our paper accordingly. We have also added comparisons to related methods, such as MLCAT you mentioned, on benchmark datasets. We would like to know whether you find our response satisfactory, or if there are more questions that we could clarify. Since the rebuttal stage is coming to an end, we are more than happy to hear your comments and address any of your further concerns during the remaining time.
>
> Best,
>
> Authors

---

> ### Author Response · Authors · 2022-12-13
> **Thanks and Further Response to Reviewer zGeX**
>
> Thanks for your response and constructive comments.
>
> For the first point, we do agree with you that Ilyas et al. are not necessarily wrong, if we believe that we are defining two different kinds of robustness. Though, we suspect whether there exists such a thing that "a feature is fundamentally robust" as assumed by Ilyas et al. We remark that Ilyas et al.'s experiments do not fully support this claim, since they only considered transferability across neural networks. From our perspective and experiments, the robustness of a feature seems always dependent on the model and attacker considered. Overall, there is not necessarily a conflict between our theory and Ilyas et al's. We will revise our writing to state their difference more clear following your suggestion.
>
> Second, why normal training does not overfit is also clear from our theory. In fact, normal training can be seen as a special case of adversarial training with an attacker with a perturbation budget $\varepsilon=0$. Therefore, all useful features are robust features (because the attacker does not change its prediction), and there is actually no useful non-robust feature in normal training (we emphasize that whether a feature is non-robust or not is attacker-specific, in both our theory and Ilyas et al). Due to the lack of non-robust features, there is no chance of inducing false non-robust mapping, and thus, normal training will not likely overfit in the same dramatic way as adversarial training.
>
> Hope it could address your concerns and please let us know if there is more to clarify.

---

### Author Response · Authors · 2022-11-16
**A Summary of Paper Updates**

We thank all reviewers for your constructive comments. We have **revised our writing and formulations** according to your suggestions for better clarity. Notably, we add extensive experiments **(5 pages in total)** listed below to further support our understandings of RO and the effectiveness of our proposed method BoAT:

- **Appendix B.2 (new)**: add experiments for RO across **different decaying epochs for LR**, which further justify the phenomenon "RO happens after LR decays".
- **Appendix B.3**: add verifications for our understandings with **different LR decay schedules** (cosine & linear).
- **Appendix B.6 (new)**: add results on the **robust generalization gap**.
- **Appendix C.2 (new)**: **improve the natural accuracy** of BoAT with a combination with knowledge distillation techniques.
- **Appendix C.3 (new)**: show the effectiveness of BoAT across **different LR schedules** (cosine & linear).
- **Appendix C.4 (new)**: add results that select best checkpoints using a **held-out validation set**, where our BoAT is still consistently superior than others.
- **Appendix C.5 (new)**: show the effectiveness of BoAT across **different types of networks** (AlexNet & MobileNetV2).
- **Appendix C.6 (new)**: compare the **training efficiency** among baseline methods, where our BoAT has little computation overhead.
- **Appendix C.7 (new)**: add comparison to a related work **MLCAT** (Yu et al. ICML 2022), where our method is also significantly better.
- **Appendix D (new)**: **quantitative visualization** of adversarial examples generated in our experiments.

---

### Decision · Program_Chairs · 2023-01-20

**Decision:**

Reject

**Justification For Why Not Higher Score:**

Please see above

**Justification For Why Not Lower Score:**

NA

**Metareview: Summary, Strengths And Weaknesses:**

Even though the authors addressed several concerns raised by the reviewers, reviewers remained skeptical of several claims. Overall, this is an interesting contribution, but the authors are strongly encouraged to follow the suggestions below towards improving the paper. Repeated/summarized from the reviews:

1) The overall pitch in the Intro and Section 2 should be changed in a future version of the paper. Papers that attempt to explain  robust overfitting, or even just overfitting.

2) Along the same lines, spending more effort to explain circumstances/conditions under which normal training does not overfit, but adversarial does, would be important. This would better motivate the study of the proposed method in the adversarial, rather than standard, setting.

3) The relationship between LR decay, dynamic features, and the proposed algorithm is still unclear.  Strengthening the connection between BoAT and dynamic features is also a good idea, and so would be reporting the result under validation-based early stopping in the main paper.